# Spatial structure, chemotaxis and quorum sensing shape bacterial biomass accumulation in complex porous media

David Scheidweiler ⊙[1] ✉, Ankur Deep Bordoloi ⊙[1], Wenqiao Jiao[1], Vladimir Sentchilo[2], Monica Bollani[3], Audam Chhun ⊙[2], Philipp Engel ⊙[2] & Pietro de Anna ⊙[1] ✉

Biological tissues, sediments, or engineered systems are spatially structured media with a tortuous and porous structure that host the flow of fluids. Such complex environments can influence the spatial and temporal colonization patterns of bacteria by controlling the transport of individual bacterial cells, the availability of resources, and the distribution of chemical signals for communication. Yet, due to the multi-scale structure of these complex systems, it is hard to assess how different biotic and abiotic properties work together to control the accumulation of bacterial biomass. Here, we explore how flow-mediated interactions allow the gut commensal *Escherichia coli* to colonize a porous structure that is composed of heterogenous dead-end pores (DEPs) and connecting percolating channels, i.e. transmitting pores (TPs), mimicking the structured surface of mammalian guts. We find that in presence of flow, gradients of the quorum sensing (QS) signaling molecule autoinducer-2 (AI-2) promote *E. coli* chemotactic accumulation in the DEPs. In this crowded environment, the combination of growth and cell-to-cell collision favors the development of suspended bacterial aggregates. This results in hot-spots of resource consumption, which, upon resource limitation, triggers the mechanical evasion of biomass from nutrients and oxygen depleted DEPs. Our findings demonstrate that microscale medium structure and complex flow coupled with bacterial quorum sensing and chemotaxis control the heterogenous accumulation of bacterial biomass in a spatially structured environment, such as villi and crypts in the gut or in tortuous pores within soil and filters.

Bacteria typically inhabit spatially complex environments, ranging from biological organs and tissues such as the gut, to soil and sediments, where resources, chemical stimuli, and physical constraints are patchy and transient[1–3]. In these environments, fluids are forced to flow within pore spaces confined between irregular surfaces. Several features characterize such confined systems, including their material composition[4,5], pore-size, pore-morphology, and permeability to fluids[6]. Spatial variability (referred to here as, heterogeneity) in these properties appears in most realistic scenarios. The gut environment, for instance, is known for its complex spatial heterogeneity, ranging across multiple scales - from microns (e.g. microvilli), hundreds of microns (e.g., villi and crypts) to several millimeters (e.g. circular folds)

[1]Institute of Earth Sciences, University of Lausanne, CH-1015 Lausanne, Switzerland. [2]Department of Fundamental Microbiology, University of Lausanne, CH-1015 Lausanne, Switzerland. [3]IFN-CNR, L-NESS, Via Anzani 42, 22100 Como, Italy. ✉e-mail: david.scheidweiler@gmail.com; pietro.deanna@unil.ch

and meters (small to large intestine), leading to complex transport of fluid, chemical gradients, and spatial organization of microbial communities[2]. Yet, how the heterogeneity of such environments determines bacterial colonization, transport of individual bacteria[7,8], cell-to-cell interactions[9], and growth[10], remains poorly understood.

Recent research suggests that the spatial arrangement of bacteria in the gut is a critical factor influencing their function and interaction with the host[2,11,12]. Several studies have demonstrated that the gut microbiota exhibits intricate spatial patterns, with distinct bacterial species occupying specific niches within the gut lumen and mucosal surface (i.e. villi and crypts)[13,14]. These patterns are influenced by various factors, including flow dynamics, nutrient availability, and ecological interactions among bacterial species[5,15]. However, studying the impact of these factors in vivo comes with obvious constraints. Therefore, researchers adopted alternative strategies such as microfluidics[16,17], organoids[18,19], and in-silico approaches[20,21] to study the impact of flow on gut related microorganisms. The results of these studies indicate that fluid movement in the gut can significantly affect bacterial behavior, including their spatial organization and interactions with the host.

Structural properties of heterogenous environments control the flow of fluids (i.e. advection)[22,23], transport of suspensions (e.g. bacteria)[7,24], solutes (e.g. nutrients) and the establishment of their gradients[25]. By combining sensory receptors and motility systems, bacteria direct their motion towards or away from gradients in dissolved oxygen, carbon sources or self-secreted metabolites[26–28]. This behavior, known as chemotaxis, entails a population benefit for bacteria at the cost of resources invested in motility[8,29,30]. Earlier research showed that chemotactic motility influences bacterial dispersal in presence of stationary[19–21], and dynamic gradient conditions[10]. However, in both scenarios, the effects of chemotaxis are limited to relatively short periods of time (up to tens of minutes), explained by diffusive dissipation of chemical gradients[31,32]. Observations on persistent chemotactic gradients, maintained in porous medium over several hours have been reported in the absence of flow when bacteria consume nutrient that is a chemoattractant itself[33]. Here we consider the scenario where the medium structure, together with bacteria transport, aggregation and chemoattractant production sustain gradients persistently in presence of flow.

Bacteria can sense concentration and gradients of self-secreted metabolites. This sensory ability allows them to detect differences in population density and coordinate gene regulation: a mechanism called quorum sensing (QS)[34]. QS is initiated by the secretion of signaling molecules, known as AutoInducers (AI), which accumulate in an extracellular environment along with the increasing cell density. This cell-to-cell "communication" mechanism enables bacteria to modulate collective behaviors, such as secretion of virulence factors, motility, and biofilm formation[35–38]. The latter leads to the transition of bacteria from the planktonic state to multicellular architectures embedded within a matrix of extracellular polymeric substances (EPS)[39,40] that allow them to persist in diverse environments[2,13]. While certain signaling systems are species-specific (known as QS type-1; such as for *Vibrio harveyi*), other signaling systems enable interspecies communication (known as QS type-2)[35]. This is the case of the mammalian gut commensal *Escherichia coli*, whose cells produce, detect and uptake interspecies signaling molecule AutoInducer-2 (AI-2). QS type-2 allows different species to sense their mutual abundance as well as interfere with neighboring cells[35]. Additionally, the regulation of QS type-2 can be affected by nutrient availability and cellular metabolism. In presence of glucose, for example, the carbon catabolite repression inhibits the AI-2 uptake of *E. coli*, therefore hindering QS induction[36,41,42].

The physical structure of the environment controls the spatial and temporal availability of resources and AIs by modulating its local transport[9]. On the one hand, flow can disrupt QS by displacing the produced signaling molecules. On the other hand, stagnant regions promote the AIs accumulation, controlling the local QS induction, as shown for of *Staphylococcus aureus* and *Vibrio cholerae*[43]. Moreover, several studies demonstrated that *E. coli*, can sense AI-2 through the Tsr chemoreceptor of L-Serine, and the regulator of the flagellar motor CheY, allowing cells to swim towards gradients of the signaling molecules[31,32]. These findings suggest that the interplay between physical structure, flow, and sensing regulation, might have remarkable implications on the bacterial colonization in complex environments.

In this study, we investigate how the interplay of the spatial structure, fluid flow, chemotaxis and QS controls the colonization and biomass accumulation within a porous micro-environment. To this end, we use a microfluidic model featuring a structure that comprise Dead-End Pores (DEPs) connected to a network of Transmitting Pores (TPs)[23], typical of spatially complex systems, such as biological tissues or soil. We show that, in such complex structures, QS is used by *E. coli* to turn stagnant DEPs into hot-spots of bacterial colonization. We demonstrate three primary mechanisms that control the overall colonization process. During the early times, accumulation of individual cells within the DEPs is driven by chemotaxis towards self-produced AI-2. Then, cell aggregation in DEPs is further promoted by motility towards gradients of AI-2 released by bacterial clusters and by cell-to-cell collision. At late times, resources limitation and accumulation of AI-2 lead to enhanced biomass growth induced by QS, which mechanically force daughter cells out of resource limited zones. This observation is explained by the persistence of self-produced AI-2 gradients for extended time, tens of hours, at the DEP-TP interface. Our results indicate that the spatial organization of complex structures plays a pivotal role in local colonization of *E. coli*.

## Results

### A microfluidic set-up to study *E. coli* colonization of a heterogeneous porous system

Within complex physical structures, the hydrodynamic competition between advection dominated pores (controlled by fluid displacement) and stagnant zones (controlled by molecular diffusion) results in heterogeneous transport conditions for solutes and microorganisms. To explore the role of microscale structures on the invasion and colonization of the mammalian gut commensal bacterium *Escherichia coli* MG1655, we designed a porous micromodel (Fig. 1a, c) comprising cavity-like irregular structures resembling those commonly found in the gut (Fig. 1d)[2,44]. With this micromodel as a template, we built Polydimethylsiloxane (PDMS) based microfluidic devices featuring cavity structures (see Fig. 1c). These cavities, referred to here as the Dead-End Pores (DEPs), are connected to a network of percolating channels, called the Transmitting Pores (TPs). We distinguish these two pore-features for the entire system (see "Methods"); a representative section is shown in Fig. 1e.

For these experiments, we use wild-type *E. coli* strain MG1655 (WT) that can produce AI-2, is chemotactic towards AI-2 gradients and is capable of QS. Each experiment is accompanied by an independent experiment wherein the WT strain is replaced by a mutant (*E. coli* Δ*luxS*) lacking the AI-2 synthase LuxS, does not produce AI-2, is not subject of QS, yet it remains motile and chemotactic.

As shown in Fig. 1b, first, we saturate the device with a motility buffer (10 mM potassium phosphate, 0.1 mM EDTA, 10 mM lactate, 1 mM methionine, pH 7.0), followed by a sharp injection (see[23]) of bacterial suspension. At this stage, the bacteria are suspended in the motility buffer, which allows them to swim (propelled by their flagella), but not to divide and grow. We set the bacteria injection flow rate at $Q = 0.1\,\mu L/min$, such that the average, Darcy, velocity in the porous medium is comparable to the measured average swimming speed of this *E. coli* strain (Supplementary Fig. S1), and to the mean fluid velocity in human gut ($20\,\mu m/sec$)[16]. Since the local fluid displacement within

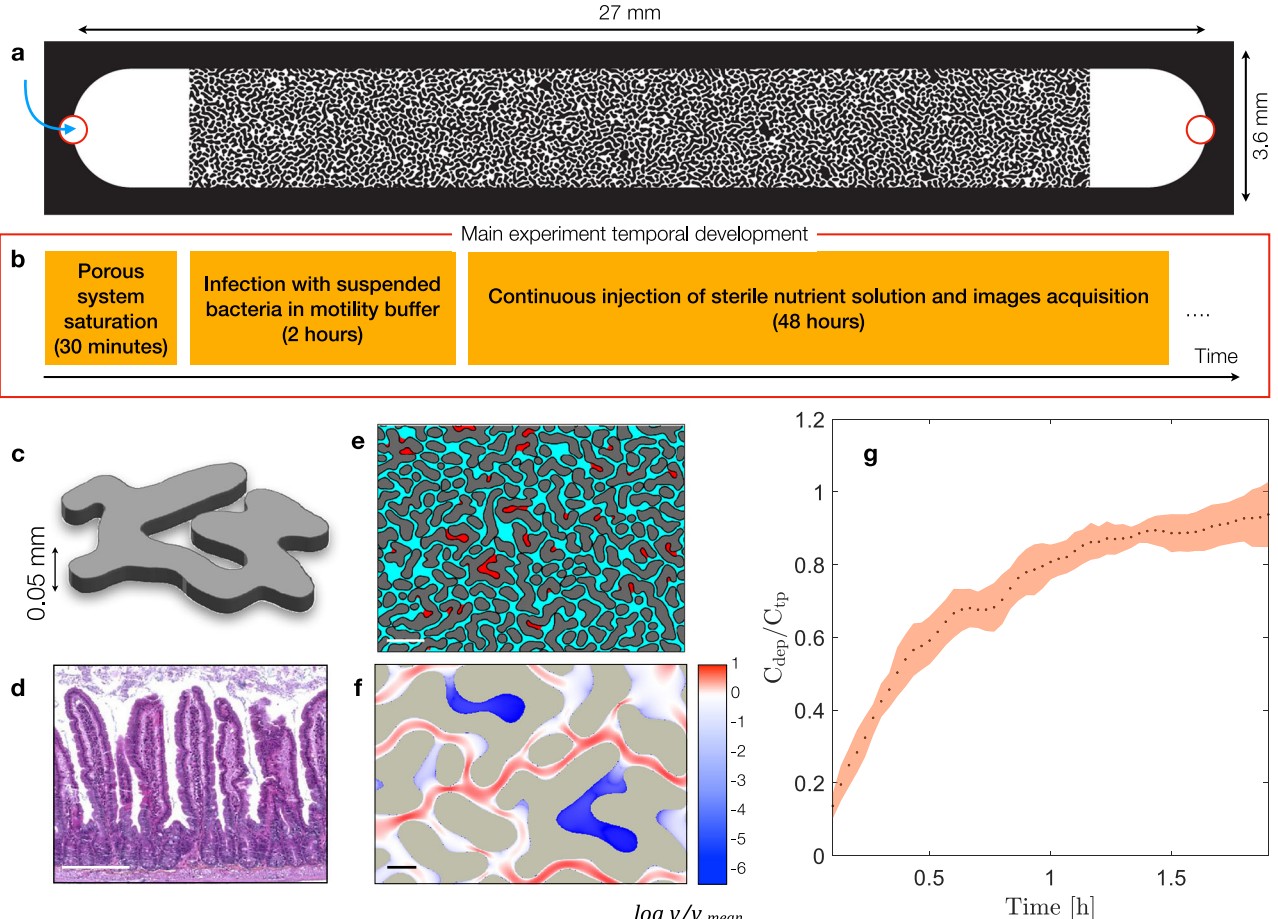

**Fig. 1 | Microfluidic device design, geometry, flow and transport characterization. a** A porous medium model microfluidics chip overview. **b** Microfluidic chip experiment time-scale. **c** Representation of one single impermeable obstacle. **d** Light microscopy image which highlights the complex geometry of mice intestine (Hematoxylin and Eosin stained). Scale bar 200 μm. Courtesy of D. Hardy, Histopathology center, Institut Pasteur. **e** Geometrical discretization into transmitting pores (TPs, cyan) and dead-end pores (DEP, red). Scale bar 200 μm. **f** Velocity field modulus, normalized by its own averaged value and shown in logarithmic scale, derived from the computed Stokes flow solutions in a subsection of the microfluidic geometry used in the experiments. Scale bar 50 μm. **g** Dynamics of porous medium colonization: retention curve representing the ratio between the concentration of bacteria in the dead-end pores ($C_{dep}$) and their concentration in the transmitting pores ($C_{tp}$) over 2 h after cell injection. Dots represent averages of independent experiments ($n = 3$); shaded areas represent standard deviation.

DEP is too low to be detected experimentally (see Supplementary Fig. S2), we compute the local fluid velocity within the porous medium by numerically solving the two-dimensional steady state incompressible Stokes flow equations in a subsection of the microfluidics geometry (see Methods). The local fluid velocity spans over several order of magnitude across TPs and DEPs: a portion of the computed velocity field modulus is shown with a logarithmic scale in (Fig. 1f). We also measure the bacterial swimming speed in a separate experiment under no-flow condition and find it to be indistinguishable between the two used strains (see Supplementary Fig. S1). With these experimental conditions, it takes approximately 2 hours (corresponding to the injection of 5 times the microfluidics volume) for suspended *E. coli* cells to invade and homogeneously fill the entire pore space. The macroscopic retention curve in Fig. 1g shows that the ratio of cell concentration, averaged over three replicas, within the two pore features DEP and TP ($C_{dep}/C_{tp}$) approaches unity at approximately 5 pore volumes (2 h). The shaded area represents the standard deviation among replicas. Once the medium is homogeneously filled with suspended cells, we switch to the continuous injection of a sterile nutrient solution (see Methods). In previous studies the biomass growth under macroscopic imposed pressure conditions reported the catastrophic disruption of the flow as consequence of the reduction of the space available to fluid flow due to cellular division[45,46]. Here, as we impose a constant flow rate, we do not observe such phenomenon.

## *E. coli* biomass distribution in complex structures

To study the role of transport on chemotaxis and QS in complex landscape, we followed the biomass accumulation of the *E. coli* strain MG1655 (WT) contrasted with its Δ*luxS* mutant (thus, AI-2 negative). In each experiment replicated 3 times, the microfluidics, initially colonized homogenously (see above), was continuously perfused for 50 hours with a sterile M9 minimal medium (MM) supplemented with 5 mM glucose as carbon source (that is a concentration comparable with that in mammalian gut[47]). Through the entire experimental time, the WT strain (as opposed to its Δ*luxS* mutant) exhibits systematically higher (up to five times exceed) biomass within the DEPs compared to the TPs (Fig. 2a, c, Supplementary movie 1 and 2). Consequently, 35 ± 1.6% of the system's total WT biomass was partitioned to DEPs whereas that of the Δ*luxS* mutant (8.1 ± 0.6 %) mirrored the volumetric proportion of the DEPs (8%) (Fig. 2b), thus unbiased. As quantitatively captured in Fig. 2b, c, such enhanced cell accumulation clearly demonstrates that the DEPs act as "hot-spots" for microbial accumulation and growth for WT but not for the Δ*luxS*.

Certainly, cells are expected to detach from the TPs due to fluid shear, the average shear rate is $u_m/h = 0.4 s^{-1}$ ($h$ being the chip thickness), where mean fluid velocity is $u_m \sim 20 \mu m/s$. Both WT and Δ*luxS* strains show similar ability to persist advection-induced shear resulting in comparable bacteria concentrations in the TPs (Fig. 2b, c). Moreover, experiments with higher glucose concentration in the medium

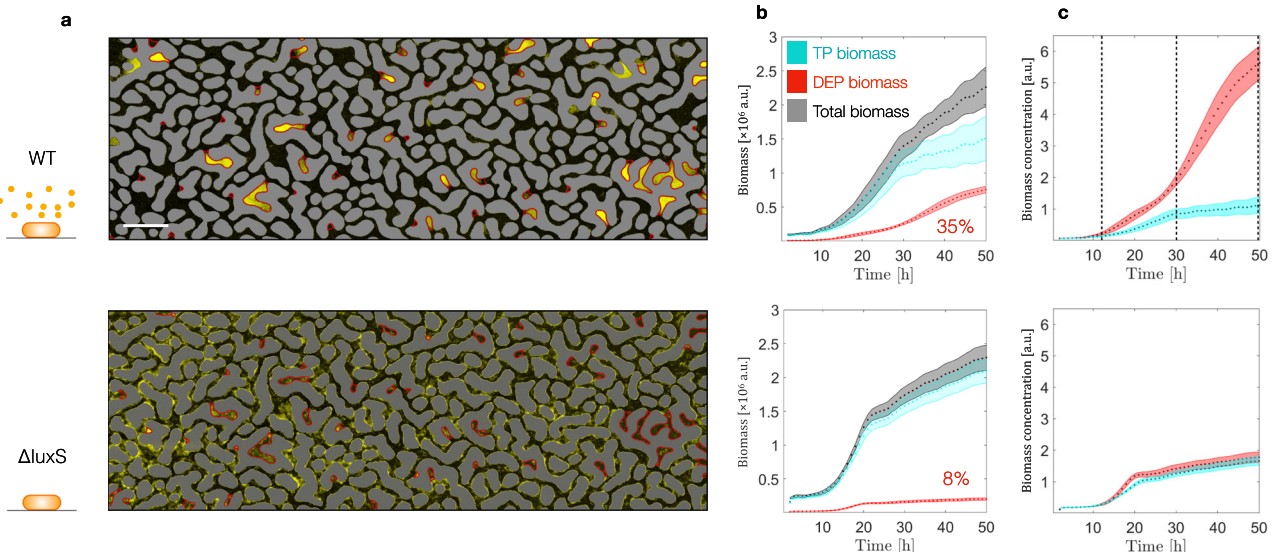

**Fig. 2 | Macroscopic biomass accumulation, for *E. coli* WT (top panels), and *E. coli ΔluxS* (bottom panels). a** Overview of microfluidic chip section depicting biased biomass accumulation through the microfluidic geometry at the end of the experiment (50 h). Scale bar 200 μm. **b** Dynamics of biomass accumulation in the TP (cyan), in the DEP (red), and in the overall geometry (gray). Dots represent averages of independent experiments ($n = 3$); shaded areas represent standard deviation. **c** Evolution of the biomass concentration in the TP (cyan), in the DEP (red) over time. Vertical dashed lines distinguish the three proposed phases: (i) individual cell accumulation in DEPs; (ii) differentiation into sessile colonies; (iii) enhanced colonies growth. Dots represent averages of independent experiments ($n = 3$); shaded areas represent standard deviation.

(50 mM; SI Fig. 3) show that the biomass of WT still accumulates in the DEPs, but it reached a higher carrying capacity only in TPs. These observations suggest that mechanisms other than flow-induced shear promote the observed higher accumulation of WT biomass in the DEP. We argue that the difference between WT and *ΔluxS* behavior is due to multiple biological activities related to chemotaxis and QS that only the former strain is capable of. We identify three distinct phases (see dashed lines Fig. 2c) during the porous medium colonization by WT, characterized by the differences in spatial distributions of biomass: (i) individual cell accumulation in DEPs, (ii) differentiation into sessile colonies, (iii) enhanced colonies growth. We described these phases in further detail below and we elucidate the underlying biotic and abiotic factors at play.

**Early times−Chemotaxis towards AI-2 promotes biomass accumulation in DEPs**

During the first 12 h of nutrient injection, we observe that WT cells preferentially accumulate in DEPs (Fig. 3a), resulting in more than 2.5-fold higher concentration of suspended cells (Fig. 3b, Supplementary movie 3 and 4). In contrast, *ΔluxS* cells disperse homogeneously across both TPs and DEPs. We suggest that this discrepancy is not the result of a different motility behavior, as WT and *ΔluxS* show the same velocity distribution (Supplementary Fig S1). Instead, we hypothesize that higher concentration of WT cells in DEPs is controlled by chemotaxis towards the self-secreted AI-2, which accumulates within the stagnant DEPs, while being advected along TPs. It is important to note that in presence of glucose, WT cells keep producing AI-2 while catabolite repression inhibits AI-2 uptake from the extracellular environment[36,41,42]. Therefore, we expect cells to continuously produce AI-2 until glucose is depleted from the pore space. While advection removes the AI-2 molecules from the TPs, they accumulate in the DEPs, thus leading to AI-2 gradients at the entrance of each DEP that, in turn, trigger chemotactic migration of WT (but not *ΔluxS*) towards the DEPs.

We examine the persistence of solute gradients within DEP-entrance and the consequent bacterial chemotaxis via two separated experiments. In the first, we quantify transport of a solute visualizing the variation in intensity of an aqueous fluorescent dye solution (Fluorescent Sodium Salt, Merck), initially saturating the same microfluidics and, then, displaced by deionized water. Fluorescent dye concentration gradients at DEPs entrance can be detected even after eluting about 1 pore volume which corresponds to 24 minutes (Fig. 3c). Thus, the gradients of this passive tracer persist for much longer than the diffusive time scale over the average DEP depth $L = 0.2$ mm, which is $T_d = L^2/D$ about 1 minute (D = 0.0006 mm²/s is the measured fluorescent dye diffusion coefficient[48]). This strong persistence results from the coupling between the flow kinematics[49,50] and molecular diffusion[48]. In our experiments with bacteria, local gradients across DEPs (where AI-2 is continuously produced by resident cells) and TPs (where AI-2 is continuously removed by advection) can persist much longer than for a passive fluorescent tracer, as long as AI-2 is produced and not uptaken.

In a separate transport experiment, we examine if these small scale and persistent gradients are sufficient to trigger the chemotactic motion by *E. coli* cells. We saturate the microfluidic device with a motility buffer supplemented with L-serine (10 mM), an amino acid toward which *E. coli* is chemotactically attracted via the same chemoreceptor (Tsr) which is used to sense AI-2[31]. Next, we displace the resident solution by a sharp front injection of a mixture of the motile WT and the non-motile *ΔfliM* cells (1:1 ratio), suspended in the motility buffer without chemoattractant. We use L-serine instead of AI-2 to separate transport and chemotaxis from other effects triggered by the AI-2 availability, such as the induction of auto-aggregation genes[38]. The non-motile strain *ΔfliM* functions as a transport control. In presence of the chemoattractant L-serine, after eluting one pore volume (24 min) the motile cells invade the DEPs 40 % more frequently (CDep / CTP = 1) than in the absence of the chemoattractant (CDep/CTP = 0.6). Their abundance quickly reaches a plateau as shown by the retention curve, quantified as the ratio between bacterial availability per pore feature (Fig. 3d). These two experiments show that the medium structure promotes gradients of chemoattractant along the DEPs that can be sensed by bacteria and attract them towards higher concentration (Fig. 3e). These results explain the motile bacteria continuous accumulation within DEPs during the first hours of nutrient solution injection.

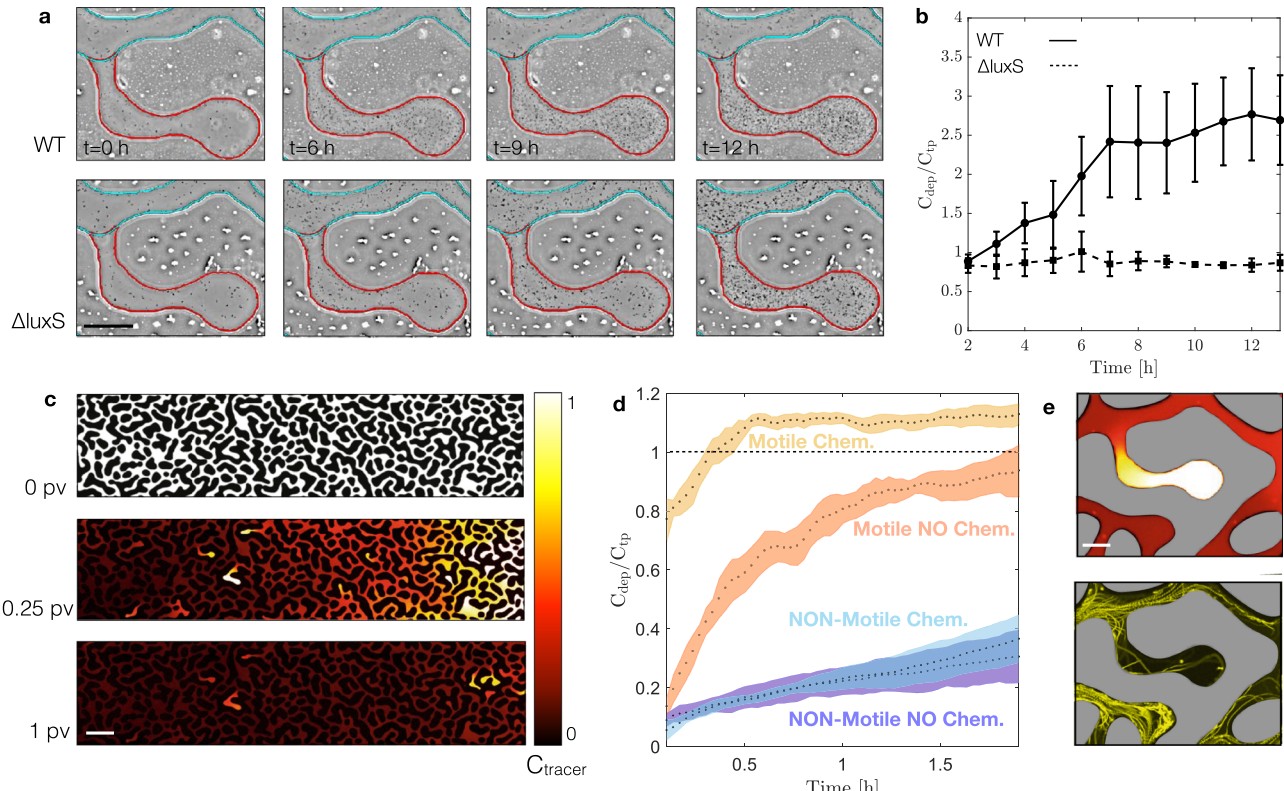

**Fig. 3 | Transport mediated accumulation of *E. coli* at early times (0-12 hours).** **a** Images at different times representing bacterial abundance in a single DEP, highlighted in red, and the nearest TP, highlighted in cyan, for *E. coli* WT (top panels), and *E. coli* Δ*luxS* (bottom panels). Scale bar 50 μm. **b** Retention curves representing the ratio between the concentration of bacteria in the dead-end pores ($C_{dep}$) and their concentration in the transmitting pores ($C_{tp}$) at different times, for *E. coli* WT (straight line), and *E. coli* Δ*luxS* (dashed line), (repeated measures ANOVA, $p < 0.0001$). Dots and squares represent averages of independent experiments ($n = 3$); bars represent standard deviation. **c** Normalized concentration map of fluorescein sodium salt at different times while displaced by de-ionized water. Scale bar 200 μm. **d** Retention curve representing the ratio between the concentration of bacteria in the dead-end pores ($C_{dep}$) and their concentration in the transmitting pores ($C_{tp}$), at different times. Experiments performed with *E. coli* WT (dark orange) and *E. coli* Δ*flim* (dark blue), in the absence of a chemoattractant. Experiments performed with *E. coli* WT (light orange) and *E. coli* Δ*flim* (light blue), in presence of a chemoattractant. Dots represent averages of independent experiments ($n = 3$); shaded areas represent standard deviation. **e** Gradient of the fluorescent tracer (top panel) and trajectories of *E. coli* WT (from time-lapse imaging) attracted towards a chemoattractant, after 1 PV (24 min). Scale bar 50 μm.

## Intermediate times—Transport and chemotaxis determine *E. coli* local architectural differentiation into sessile and suspended aggregates

At times larger than 12 h, we observe that cells within the DEPs start forming aggregates. Similar observations in earlier studies have been attributed aggregates formation to chemotaxis towards AI-2 that promotes cell-to-cell collisions in *E. coli*, a process mediated by cells encounter and aggregating factors as outer membrane adhesins, fimbriae, pili or curli fibers[31,32]. Current understanding of this mechanism is limited to relatively simple geometries without flow, where gradients of self-produced signaling AI-2 molecules persist for a short period (i.e. tens of minutes[32]), before diffusion mixes AI-2, dissipating its gradients with subsequent loss of chemotaxis. This results in disaggregation, because motile cells detaching from aggregates are higher in number than those that join them[32]. We argue that in complex landscapes, as in our experiments, the persistence of AI-2 gradients is sustained by the combined effect of advection, constantly removing AI-2 molecules from TPs, and production of AI-2 by cells trapped within DEPs. Therefore, motility towards such gradients induces aggregate formation without exhibiting disaggregation, up to about 30 h, as visible in Fig. 4a.

We further show that not only biomass accumulates differently across the two strains, but also the architectural differentiation is pore-class dependent. In the TPs, the WT strain forms a homogenous layer of cells and reaches its carrying capacity at around 30 hours, while in the DEPs cell aggregates continue to increase in size until the entire DEP space is occupied (Fig. 4a, b). In contrast, the Δ*luxS* strain forms sessile colonies, which are architecturally the same in TP and DEP (Fig. 4a, b). This is captured in the aggregate-averaged mass distribution (probability density function, PDF) within the DEP that stays nearly constant for Δ*luxS* but keeps shifting towards larger values for WT (Fig. 4b), thus implying that the aggregate size increase in DEPs stems from a combination of binary cell division and recruitment of new cells via chemotaxis. Interestingly, increasing concentration of glucose by 10 folds in growth medium did not affect WT biomass accumulation in DEPs as opposed to TPs, the latter exhibiting higher cell-carrying capacity. We explain this by arguing that nutrient consumption by resident cells could not be balanced by diffusive transport of the resource in DEP. Thus, only TP exhibited higher carrying capacity (Supplementary Fig. S3), as the injected solution carries fresh nutrient. Eventually, due to chemotaxis and cells aggregation, the pore space becomes crowded and the overall biomass accumulation slows down.

## Late times—Nutrients limitation and Quorum Sensing control enhanced biomass accumulation in DEPs

After 30 h of colonization, WT biomass accumulation in the TPs slows down and almost stops (see Fig. 2b and Supplementary movie 1). In contrast, in the DEPs biomass accumulates faster than in the intermediate phase (12–30 h), until the growing biomass extrudes from an

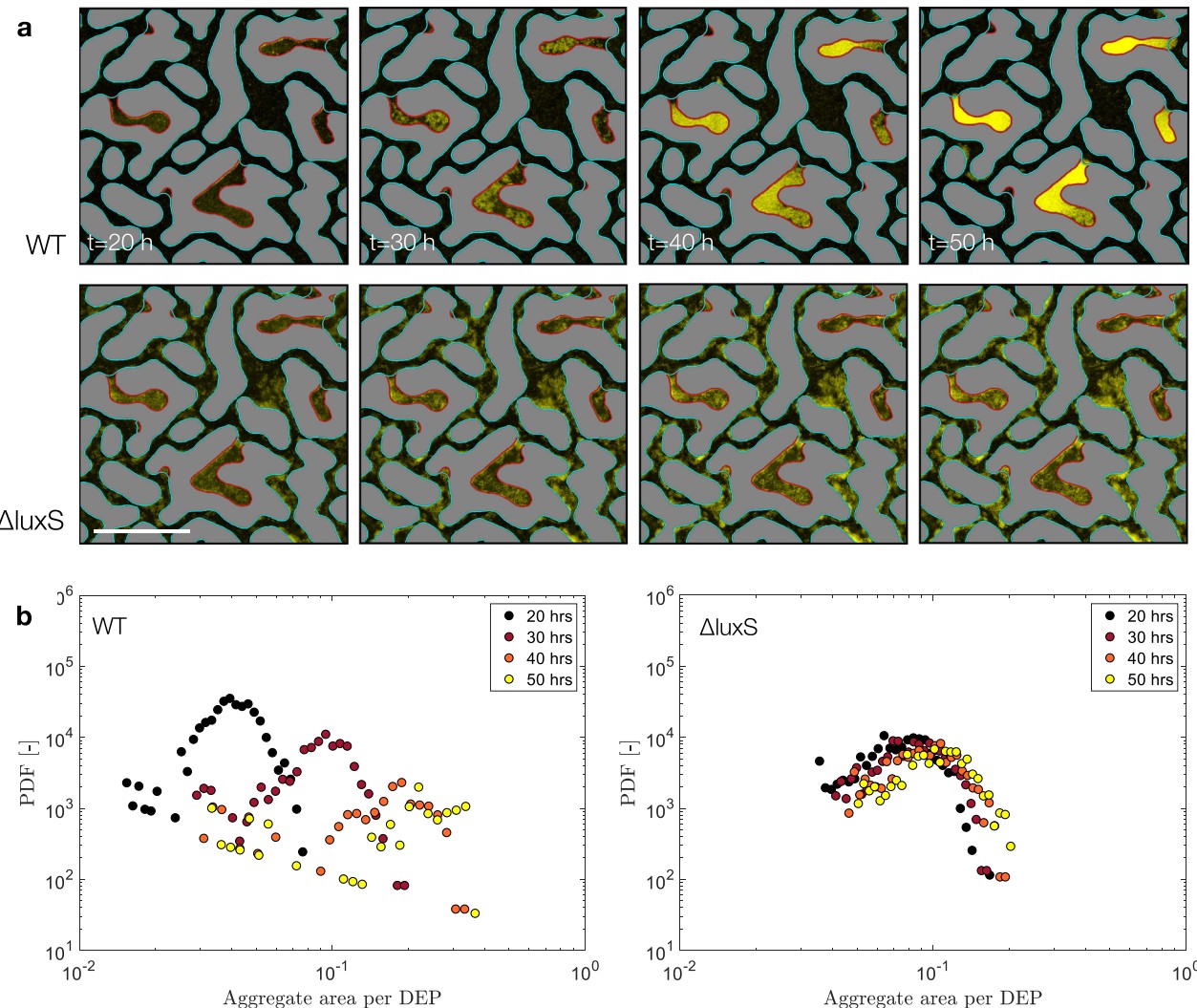

**Fig. 4 | Cell aggregates size dynamics of *E. coli* in presence or absence of AI-2 in DEPs over time. a** DEP's colony growth of *E. coli* WT (top panels), and *E. coli* Δ*luxS* (bottom panels) at different times. Cells are shown in yellow over a dark background, scale bar 200 μm. **b** Probability density function (PDF) of individual colony biomass per colony surface, at different times, for *E. coli* WT (left panel), (repeated measures ANOVA, $p < 0.0001$), and *E. coli* Δ*luxS* (right panel), (repeated measures ANOVA, $p > 0.1$). Dots represent average PDF of independent experiments ($n = 3$).

individual DEP into the nearest TP (Fig. 5a, and Supplementary movie 1). We, therefore, hypothesize that around 30 h high cell densities and high levels of AI-2 induce QS and activate gene expression that controls enhanced biomass accumulation, a QS regulation known to occur in *E. coli*[38,51]. In this regard, the gene *lsrR* plays an important role for *E. coli* by directly controlling AI-2 uptake[52] and regulating the expression of several important biofilm-related genes such as *wza*, involved in the synthesis of colanic acid - a building block of EPS, and *flu*, which encodes the proteins that mediates cell-to-cell aggregation[32,33].

To test if the observed biomass accumulation is the result of QS, we estimated from the *lsrR* activity when and where *E. coli* cells start to internalize AI-2, and if there is a temporal agreement with the quantified TP and DEP biomass accumulation dynamics. To this end, we perform a set of three independent experiments where WT *E. coli*, equipped with a *lsrR* gene promoter-*gfp* reporter (pMSs201-P$_{lsrR}$-gfp)[53], was followed in the same microfluidic chip and under the same flow conditions as described above. GFP signal remained below detection limits during the first 25 h, however, between 25 and 30 h, the *lsrR* expression (GFP fluorescence normalized per unit of biomass) sharply increased until the end of the experiment, notably, in DEPs but not in TP (Fig. 5b, c). From this we concluded, that the hypothesized

AI-2-driven onset of QS is taking place in DEPs (but not in TPs) and it is temporarily coinciding with the observed peak of biomass accumulation in the DEPs.

It has been reported that phosphoenolpyruvate (PEP)–dependent sugar phosphotransferase system (PTS) modulates QS: in presence of glucose (or glycerol), carbon catabolite repression and direct binding of LsrK to phosphocarrier protein (HPr) inhibits AI-2 uptake and phosphorylation, therefore hindering *lsr* operon expression[36,41,42] and QS regulatory cascade. We argue that, in the DEPs (but not in the TPs), despite constant biomass and concordant AI-2 accumulation (as we described above), QS remains repressed as long as glucose is available and it becomes induced upon glucose depletion. To test this, we measured P$_{lsrR}$-gfp reporter activity along the depth of all DEPs (averaged along the skeleton of each individual DEP structure; see Methods). Figure 5d (solid lines, color coded by time) shows the reporter activity along each DEP and averaged over all DEPs: the signal is strong deep down each DEP, and it becomes progressively weaker towards the entrance (at the juncture with the adjacent TP). The transition between high to low QS reporter activity takes place at a distance ζ away from the DEP-TP juncture, which depicts a zone where AI-2 is internalized by cells, and where glucose is depleted. Local glucose depletion results from the competition between glucose consumption

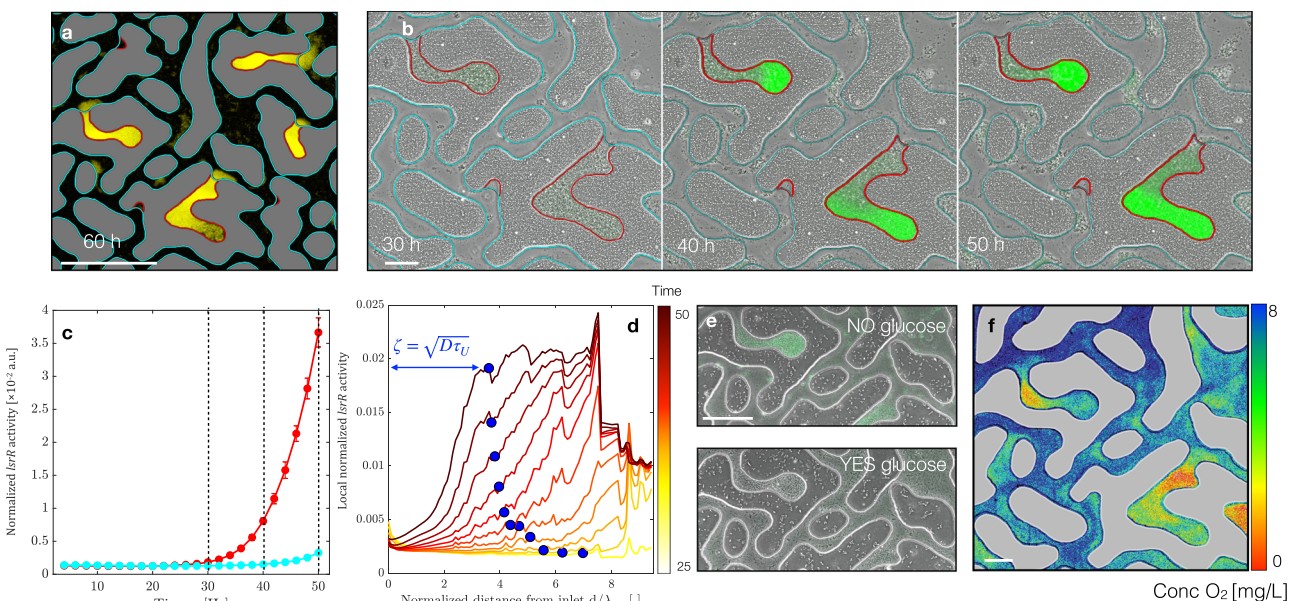

**Fig. 5 | QS induction of microbial growth in cell aggregates within DEPs.**
**a** Biomass extruding out of the DEPs after 60 hours. Scale bar 200 μm. **b** Images
showing the biomass accumulation in one DEP (phase contrast) and the $P_{lsrR}$ activity
(green superimposed color) of *E. coli* WT (pMSs201-$P_{lsrR}$-gfp, the reporter for AI-2
re-uptake/QS) at different times. Scale bar 50 μm. **c** Quantification of the $P_{lsrR}$
activity, normalized by biomass abundance for *E. coli* WT, (repeated measures
ANOVA, $p < 0.0001$) in TP, cyan, and DEP, red. Dots represent averages over all
pores ($n = 230$); bars represent standard deviation. **d** Averaged normalized $P_{lsrR}$
activity along DEP length over 25–50 h period. Glucose limitation length scale, $\zeta$, is
represented with a blue circle. **e** Images representing the biomass accumulation in a
single DEP superposed to the $P_{lsrR}$ activity (green) of *E. coli* WT, in presence of
0.1xLB without glucose ($n = 3$, top panel), and 0.1xLB amended with 5 mM of glu-
cose ($n = 3$, bottom panel). Scale bar 100 μm. **f** Map of the oxygen concentration
resulting from the coupling of advective-diffusive transport and *E. coli* WT uptake
after 50 h of experiment. Scale bar 50 μm.

by microbial cells and its diffusion from the nearest TP and occurring
through the cell biomass itself. We could identify high glucose-
consumption region at a short distance from the DEP entrance using
oxygen sensor (Fig. 5f). Accordingly, deeper in the DEP, the glucose
availability (as depicted by oxygen consumption) declines. On the one
hand, the characteristic time $\tau_U$ for a given biomass to consume glu-
cose can be estimated knowing the DEP-averaged biomass estimated
from each image (see Methods). On the other hand, the characteristic
time for glucose to diffuse across the length scale $\zeta$ is $\tau_D = \zeta^2/D$. Thus,
for a given biomass, glucose diffusion and consumption equilibrate at
the distance $\zeta$ from the biomass border, when $\tau_D = \tau_U$. In Fig. 5d (blue
circles), we represent the glucose limitation length as $\zeta = \sqrt{D\tau_U}$, where
$\tau_U$ is estimated from the average DEP biomass observed and assuming
a constant glucose concentration of 5 mM at the DEP-TP juncture. A
numerical simulation capturing the whole diffusion/consumption
dynamics of glucose confirm this prediction (see Methods and Sup-
plementary Fig. S9).

To confirm that AI-2 is not taken up in presence of glucose, we
perform two control experiments in which cells of *E. coli* (pMSs201-
$P_{lsrR}$-gfp) reporter strain are exposed to a constant flow of 10-fold
diluted LB (0.1xLB) with and without 5 mM glucose. Despite the two
conditions result in a similar biomass accumulation, we observe con-
trasting results in the reporter activity. In the presence of glucose, the
reporter activity remains below detection at all times, while without
glucose, the reporter shows higher activity in the DEPs than in the TP
(Fig. 5e and Supplementary Fig. S4). As a final remark, we believe that
our results do not depend on the specific value of the imposed flow
rate Q, as far as it is strong enough to re-fresh nutrients and oxygen
while depleted in the transmitting pores but it is low enough to be
within the laminar regime.

## Discussion
The mammalian gut commensal *E. coli* coordinates population beha-
viors via secreting, sensing, and up-taking the chemoattractant and QS

signaling molecule, AI-2[42]. In a spatially complex micro-environment,
the local availability of dissolved AI-2 is controlled by the coupling of
fluid flow in advection dominated zones connecting areas of fluid
stagnation[43,54]. In this work, we use microfluidic experiments and
automated time-lapse video-microscopy to explore the role of differ-
ent structures, Transmitting and Dead-End pores (TPs and DEPs), and
the consequent flow heterogeneity on bacterial colonization of a sys-
tem that mimics the spatial complexities in the gut[23] and other (por-
ous) environments. We show that flow heterogeneity translates into
persistent gradients of signaling molecules and carbon resources
(here, glucose) with consequences on the colonization and spatial
organization of *E. coli*. Specifically, we distinguish different mechan-
isms controlling the colonization of *E. coli*: (a) chemotaxis driven by
gradients of self-produced chemoattractant AI-2, (b) aggregation
mediated by cells collision, and (c) AI-2 uptake which regulates
enhanced growth via QS. Despite the overall amount of biomass
growth for *E. coli* WT and Δ*luxS* mutant (that do not produce AI-2) are
similar after 48 h, their spatial distribution and growth dynamics differ
significantly.

We show that production of AI-2 and its accumulation within DEPs
promote chemotactic migration towards and cells accumulation
within the DEPs that, at later times, triggers QS and enhanced microbial
growth. This pore-specific colonization does not occur in a Δ*luxS*
(lacking AI-2 synthase) mutant; therefore, this phenomenon depends
on the ability of the WT strain to produce and sense AI-2 gradients.
Taken together, these findings show that chemotaxis and QS control
the spatial segregation of bacterial strains based on the strain-specific
response to AI-2. This is an indication of an active role of AI-2 secretion
and sensing in initiating ecological niche segregation[55,56], and therefore
stable co-existence of different *E. coli* strains in spatially complex
environments.

To conclusively claim that the observed *E. coli* accumulation in
the DEP is due to the chemotactic activity toward the self-produced
AI-2, we also consider the general non-chemotactic *E. coli* strain

$\Delta cheA$ and the AI-2 receptor-deficient strain $\Delta tsr$, as further negative controls. Flow experiments performed in triplicates with these two strains, separately, display similar results as the ones described for $\Delta luxS$, in terms of early times DEP-TP individual cells accumulation (see Fig. S5a, b) and overall biomass accumulation (see Fig. S6a, b). Moreover, this is supported by results of competitive experiments with *E. coli* in murine gut, where strains characterized by different AI-2 detection capability (WT and $\Delta lsrB$) coexist by occupying different niches[57].

While chemotaxis is energetically costly, it entails several benefits such as efficient nutrients acquisition, avoidance of harmful substance and colonization of new environments[58]. We show that chemotaxis allows *E. coli* to colonize cavities such as the interspace between villi and crypts, where they likely outcompete local invaders due to their higher local population. These findings complement previous works, where has been shown that in absence of flow *E. coli* exhibits a chemotactic response towards gradients of self-secreted amino acids[59–61], or while consuming resources, migrating towards areas with higher nutrient concentration[33]. Additionally, not only concentration, but also quality of carbon may influence the bacterial chemotactic response. In *E. coli*, the expression of motility genes is inversely proportional to the growth rate achieved on different carbon sources[62]. In most conditions, glucose is the best carbon source for *E. coli*, and for this study, we chose a concentration comparable with that found in mammalian guts (5–50 mM)[47]. In presence of other resources, chemotaxis could therefore result in different colonization pattern.

Chemotaxis toward AI-2 mediates cell-to-cell aggregation until receptor saturation (Tsr) or until diffusion mixes the gradients[31,32]. In our model system, the heterogeneous flow and AI-2 production sustain AI-2 gradients over 48 h. This cell accumulation driven by chemotaxis along the gradients towards DEPs for such long times promotes growth of bacterial aggregates, whose formation is mediated by cell-to-cell collision in the confined space of a DEP and cell division. Notably, these cell agglomerates do not disaggregate after the characteristic diffusion time $l^2/D$ when diffusion is supposed to have homogenized AI-2 concentration over the colony size $l$, therefore hindering chemotaxis, as previously reported[32]. Here, we show that the medium structural heterogeneity together with fluid flow sustain aggregates formation over tens of hours making DEPs hotspots of microbial colonization.

At later times (30–50 h in our experiment), we observe that biomass accumulation within the confined space of DEPs induces QS that, in its turn, mediates enhanced biomass growth leading to the transition towards a sessile life style. Furthermore, using a fluorescent reporter of *lsrR* expression, a gene involved in the uptake of AI-2 and controlling biofilm phenotype[51,63], we conclude that QS and the transition towards sessile life style happens as a response to resources limitation. We show that AI-2 uptake begins from the bottom end of each DEPs (Fig.5b–d), presumably only once glucose is depleted and diffusion from the nearest TP is not sufficient enough to provide more glucose. To sustain this conclusion, we computed the diffusive length scale $\zeta$ that glucose travels along the DEP before becoming completely depleted by cells uptake. Our model, corroborated by numerical simulations (see Methods and Fig. S9), shows that this length scale well predicts the fluorescent signal front emitted by the *lsrR* reporter along DEP.

Evidently, once glucose is depleted from the DEP, another carbon source is needed for *E. coli*, to sustain the QS-dependent microbial growth. It is known that glucose is partially catabolized and excreted as acetate by *E. coli*, a process known as acetate overflow metabolism[64]. Only once acetate-producing carbon sources are depleted (glucose, herein), *E. coli* can re-uptake previously excreted acetate via acetyl-coA synthetase (*acs*). This mechanism, called acetate-switch[65], has been reported for *E. coli* growing within confined environment[66].

Interestingly, in *Vibrio fischeri*, a squid symbiont, this metabolic switch has been recognized to be coordinated by QS controlling *acs* expression[67,68]. This synthetase provides an overall advantage in colonizing the crypts of the squid light-organs, compared to *acs* deficient strains[67,68]. To what extent the two processes (QS and acetate-switch) are coupled in *E. coli*, remains to be understood.

*E. coli* cells uptake AI-2, however they do not use it as an energy resource[69]. This metabolic expense has been shown to return a colonization advantage by interfering with neighboring cells[35]. We propose here another advantage: QS is beneficial for bacteria to physically escape unfavorable condition, via enhanced production (and shedding) of biomass. To test this hypothesis, we equipped the microfluidic device with transparent oxygen-sensors[70] to estimate the availability of dissolved oxygen within the two pore classes. As expected, after 30 h the dissolved oxygen concentration is lower deep in the DEPs due to its consumption via cell-respiration and reduced renewal due to the tortuous structure of a DEP, which is not directly accessible to flow (Fig. 5f). Hence, the excess of biomass extrudes from DEPs towards the DEP-TP interface where oxygen and glucose concentration are higher. This is corroborated by biofilms models that incorporate reaction-diffusion effects, where it has been shown that biomass production pushes daughter cells towards a more oxygen rich environment[71–73]. Peculiarly, once the extruding biomass reaches glucose-rich environment (close to the pore entrance), it becomes metabolically more active as manifested by practically anoxic zone and QS negative (Fig. 5b, f). Our findings support the hypothesis that bacterial cells within confined environments invest their energy into QS-triggered biomass production to turn resource limited zones into hot-spots of intense activity (e. g., cell division) to mechanically escape unfavorable conditions.

We argue that, overall, the impact of medium structure, chemotaxis and quorum sensing on bacterial biomass accumulation described here takes place also for TPs of distributed size and DEPs of distributed width (and not only depth). In the first case, the flow and transport of nutrients and oxygen in each of TPs should be controlled by their own width $w$, as the local permeability is expected to vary as $w^2$ (as in a pipe). For the second case, let's consider that we defined a DEP as the structural unit of the medium for which the local maximum inscribed circle touches the solid grain structure in three points belonging to the same grain. This condition happens when the cavity within a grain is deeper than wider. In this situation, as discussed in ref. 23, the fluid flow exhibits complex laminar vortexes of exponentially decaying intensity moving inside the DEP. Thus, we expect that even for width-distributed DEP, as far as they are DEP according to our geometrical definition, the fluid flow within them is sufficiently low to sustain the bacterial accumulation and AI-2 gradient persistence at the DEP-TP juncture and, finally, the late time QS-dominated biomass production. Moreover, we argue that our results do not depend on the specific value of the imposed flow rate Q, as far as it is strong enough to replenish nutrients and oxygen while depleted in the Transmitting Pores but it is low enough to be within the laminar regime.

To conclude, understanding how bacteria colonize complex and heterogeneous environments, such as the gut, is essential for comprehending host-microbe interactions and their impact on health and disease. We show how the spatial structure and flow heterogeneity of these environments play a crucial role in shaping bacterial behavior and colonization, particularly through chemotaxis and quorum sensing. This is of great relevance, as AI-2 is a universal inter-species communication system[35], and different bacterial species can produce and respond to it, modulating gut microbiota composition[74]. We showed here how persistent chemotactic gradients and QS may affect bacterial behavior in systems characterized by complex structures for an isolate strain and, thus likely, also for multispecies gut (and other) microbial communities.

## Methods

### Bacterial strains and growth condition

Microfluidic chip experiments were performed using *E. coli* strain MG1655 and its derivatives: the motile wild type (WT) *E. coli* MG1655 (CGSC#: 8237; Yale *E. coli* genetic stock center) tagged with GFP (pME6012-$P_{tac}$-gfp)[75]; the non-flagellated mutant *E. coli* MG1655 (*ΔfliM*) tagged with mCherry (pME6012-$P_{tac}$-mCherry)[76]; the mutant lacking the AI-2 synthase gene *luxS*, *E. coli* MG1655 (*ΔluxS*) tagged with GFP (pME6012-$P_{tac}$-gfp, this work); the mutant lacking the gene *cheA*, *E. coli* MG1655 (*ΔcheA*)[77]; the mutant lacking the gene *tsr*, *E. coli* MG1655 (*Δtsr*)[77] and AI-2 reporter *E. coli* MG1655 (pMSs201-$P_{lsrR}$-gfp)[53]. The inoculum for the microfluidic chip experiments was grown overnight in 4 mL of lysogeny broth Miller (LB) with 25 μg/mL of kanamycin at 37 °C and 180 rpm shaking, followed by 1:100 dilution in fresh LB and incubation until exponential growth phase (~3 hours). Resulting cultures were harvested by centrifugation (2300 g, 5 min) and resuspended in the motility buffer. The motility buffer consisted of 10 mM potassium phosphate, 0.1 mM EDTA, 10 mM lactate, 1 mM methionine, pH 7.0[78]. Chip perfusion medium was M9 minimal medium containing per liter 64 g $Na_2HPO_4$-$7H_2O$, 15 g $KH_2PO_4$, 2.5 g NaCl, 0.24 g $MgSO_4$, 0.0015 g $CaCl_2$-$2H_2O$, supplemented with 5 mM or 50 mM glucose.

### Microfluidic devices

We designed the porous micromodel exploiting the method of solid-state dewetting, which results in a surface morphology exhibiting spinodal-like structure and a disordered hyperuniform character described in[23]. We printed the micromodel geometry into a silicon wafer via soft lithography, depositing a layer of SU-8 2150 (MicroChem Corp., Newton, MA) with controlled thickness (0.05 mm) via spin-coating. The wafer acts as a mold for liquid polydimethylsiloxane mixed with 10% by weight with its own curing agent (PDMS; Sylgard 184 Silicone Elastomer Kit, Dow Corning, Midland, MI). After solidification, we plasma-sealed the microfluidic device onto 25 mm × 75 mm glass slides.

### Microfluidic experiments

We performed the microfluidic experiments within a constant temperature microscope incubator (OKOlab) maintained at 37 °C. First, we saturated the microfluidic device with a motility buffer and subsequently injected the bacterial suspensions via two inlet-ports operated on a 3-way-valve system. This system allowed to generate a sharp front of suspended bacteria at the chip inlet, as described in[23]. We imposed a constant flow rate (Q), of 0.1 μL/min with a syringe pump (PHD-ULTRA, Harvard Apparatus) eluting one pore volume (PV, i.e. the volume of the entire porous channel) every 24 min. After 5 PV, we switched the inflow from the bacterial suspension to the sterile nutrient medium keeping the flow rate same and recorded the biomass accumulation over 2 days.

### Porous medium characterization

At the end of each experiment, we created a "mask" image to characterize the porous structure. To this end, we saturated the porous medium with a florescent dye (fluorescein sodium salt, from Merck, visible through a GFP filter cube) and recorded a large image of the entire structure. After thresholding, binarizing the mask image (pore-space: 1; solid-space: 0), we geometrically discretized the two pore classes, TPs and DEPs, using the maximum inscribed circle method described in[23]. From the binary image we obtained then the skeleton, a 1-pixel width representation of the pore space (Supplementary Fig. S7a). We assigned a segregation index (ζ) to the pore-regions based on the number of neighboring grains. Hence, a segregation index (ζ) of 1 has distinguished the dead-end pores from the transmitting pores with ζ > 1. This characterization allowed to generate a "mapped mask", where grains have been attributed a pixel value of 0, dead-end pores a pixel value of 1, and transmitting pore a pixel value of 2 (Fig. 1d, gray, red and cyan, respectively). The medium is quite homogeneous, such that the statistical distribution of λ is narrow and has a strong peak close to the mean pore-size $λ_m = 0.04$ mm (Supplementary Fig. S7b).

### Microscopy

We performed time-lapse imaging with an automated transmission light microscope (Eclipse Ti2, Nikon) equipped with a CMOS camera (Hamamatsu ORCA flash 4.0, 16-bit, 6.5 μm per pixel) and controlled by the software Elements (Nikon). This integrated system allows for automatic capture of large image (stitching 3×9 individual pictures) time series for the entire porous domain, with multichannel optical configuration. We collected individual pictures (2048 × 2048) with a Nikon objective 10X magnification (0.65 μm/pixel) for each time series in bright field, phase contrast and fluorescence optical configurations. For the fluorescence optical configuration, we used Nikon GFP HQ and mCherry HQ filters coupled with a Spectra X light engine.

### Bacteria accumulation analysis

We quantified pore-space accumulation of bacteria based on the detected pixel-intensity attenuation from the time-series of large images captured using fluorescence or bright field configuration, as in[10]. Identification of individual cells, in the transport experiment, has been performed by fluorescence microscopy, total biomass has been derived by bright field images. First, we removed the background noise by subtracting from each large image the one taken at the beginning of the experiment before cells injection. Next, we multiplied each large image with a binary mask to isolate the accumulations within the transmitting pores. We processed each dead-end pore independently and identified the boundaries of individual clusters of accumulated biomass using an intensity threshold (based on mean of regional maximum intensities). We computed the area of individual clusters to compute the local concentration distribution by taking the sum of all intensities within each cluster normalized by the respective cluster area (see Fig. 4b).

### Reporter analysis

We followed the same protocol for the images acquired in bright field but in fluorescence configuration (GFP filter cube), with fluorescence intensity being a proxy of the *lsrR* activity. We discretized between DEP and TP, and then normalized the fluorescence intensity by the intensity of the biomass acquired in bright field. We further measured local biomass abundance and reporter activity along the depth of all DEPs, by averaging GFP fluorescence and light intensity signals within the maximum inscribable disks centered along the skeleton of the individual DEP structure, every 10 pixels (Supplementary Fig. S8).

### Particle tracking

To measure motility speed of individual *E. coli* cells (both WT and *ΔluxS*), we performed an independent experiment to track trajectories of bacteria under the no-flow condition. After loading the bacteria suspension (in motility buffer) into the microfluidic channel, we insulated the inlet/outlet ports. We captured time-series images using phase-contrast configuration at a magnification of 15X and acquisition rate of 100 frames/sec with an exposure time of 2 ms for a total duration of 40 s. To reconstruct the trajectories of individual bacterial cells, we follow the same method described in[7]. The velocity probability density function (PDF) based on approximately 7500 trajectories for both bacteria strains are shown in Supplementary Fig. S1.

### Numerical simulation of flow velocity

We performed a stationary creeping flow simulation using COMSOL Multiphysics over a smaller microfluidic domain of 9.6 mm × 3.6 mm with the same structure, as in[23]. The computational resolution is high enough to ensure a divergence free velocity field.

## Consumption-diffusion of glucose in DEP

The limit $\zeta$ of the glucose depleted zone within the DEP results from the competition between glucose consumption by microbial cells and its diffusion from the nearest TP through the biomass itself. We estimated the glucose consumption time $\tau_U$ by the cells within a DEP as the uptake rate $U_{1c}$ per dry mass of *E. coli* (10 mMol per second per g of dry mass of bacteria[79]) multiplied by the DEP biomass, $c_B$ The latter is estimated by assessing, first, the average number of cells in a DEP from each image. Then, the mass of cells in DEP is given by the product of their estimated number and the dry mass of an individual cell (e.g.[80]): finally, the average glucose uptake rate in the DEPs is $U_T = U_{1c} c_B$. Then, assuming a constant glucose concentration $c_0$ of 5 mMol at the DEP-TP juncture (which is the glucose concentration in the injected solution), we get $\tau_U = c_0/U_T$. Thus, this transition between high and low glucose is supposed to superpose to the measured reporter (P$_{lsrR}$.gfp) signal along the DEP length. The dynamics of the glucose diffusion and consumption is described by mass conservation, that for the 1d system represented in Supplementary Fig. S9a reads:

$$\frac{\partial c_G(x,t)}{\partial t} = D_G \frac{\partial^2 c_G(x,t)}{\partial x^2} - U_{1c} c_B(x,t) \tag{1}$$

where $c_B$ represents the local biomass concentration (measured) and $U_{1c}$ the glucose uptake rate. We numerically solve the previous partial differential equation for glucose diffusion and consumption with a backward Euler scheme, as shown in Supplementary Fig. S9b at different times (light to dark for increasing times). We couple the numerical solution $c_G(x,t)$ to the AI-2 diffusion and uptake as proposed by[63,81]:

$$\frac{\partial c_{AI-2}(x,t)}{\partial t} = D_A \frac{\partial^2 c_{AI-2}(x,t)}{\partial x^2} + k_{A+} c_B - k_{A-} f(x,t) c_B, \tag{2}$$

The diffusion coefficient of AI-2 $D_A$ is inferred from Stokes-Einstein equation, based on the known value of $D_G$ and the values of the molecular masses of glucose and AI-2 molecules. The function $f(x,t)$ has value of 0 if the local glucose concentration is above a given threshold and 1 if it is below, while $k_{A+}$ and $k_{A-}$ are the production/uptake rates of AI-2. The numerical solution of the latter partial differential equation, shown in Supplementary Fig. S9c for different times (light to dark for increasing times), allows us to estimate the *lsrR* signal as

$$lsrR = (1 - c_{AI-2}(x,t))(1 - c_G(x,t)), \tag{3}$$

that is shown in Supplementary Fig. S9d. From this spatial profile, we numerically estimate $\zeta$ as the location from the DP entrance where the simulated *lsrR* signal is above the 80% of its own maximum (blue dots in Supplementary Fig. S9d simulated and Supplementary Fig. S9e measured). Figure S9f shows the good agreement between the simulated $\zeta$ and its estimate $\sqrt{D\tau_U}$.

## Oxygen measurement

We optically quantified the dissolved oxygen within the medium pores by means of transparent planar optode sensors built within the microfluidics[70]. The planar sensor consists of a thin layer (<3 μm) of a solid polymer matrix containing two fluorescent dyes. The first is quenched by molecular oxygen whereas the second dye provides a reference signal. We independently imaged the local fluorescent signal intensity of the two dyes by means of two customized filter cubes (450/650 nm Ex/Em and 450/520 nm Ex/Em). The oxygen map is then derived from the ratio between the signal of the two dyes.

## Statistical analysis

Repeated measures analysis of variance (ANOVA) was carried out when two or more treatment groups were compared, and two-sample t-test when comparing two groups. Statistical analysis was performed using the software Matlab. *P* values < 0.05 were considered to indicate statistical significance.

## Reporting summary

Further information on research design is available in the Nature Portfolio Reporting Summary linked to this article.

## Data availability

The collected experimental data (pictures), the source data for the plots together with codes used for their analysis and interpretation, as discussed in this study, are available in the Zenodo database https://doi.org/10.5281/zenodo.10254339.

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

## Acknowledgements

We thank Jan Roelof van der Meer and Thomas Keith Wood for providing the strains. We thank Sergey Borisov for providing the oxygen optode sensors used in this study. The work has received support from the FET-Open project NARCISO (ID: 828890) and of Swiss National Science Foundation (grant ID 200021_172587). D.S. acknowledges the Swiss National Science Foundation Grant (P500PB_211100). M.B. acknowledges the support of Agritech project (Centro Nazionale per le Tecnologie dell'Agricoltura - PNRR 2022-25).

## Author contributions

D.S., A.D.B and P.d.A. designed the research, M.B. provided the dewetted HPS samples, D.S. performed experiments, A.D.B. performed numerical simulations, D.S., A.D.B. P.E and P.d.A. analyzed the data, D.S., V.S. and A.C. prepared the mutant and reporter strains, W.J. contributed to the control experiments and D.S., A.D.B., V.S., P.E and P.d.A wrote the manuscript with contribution form all authors. D.S. and A.D.B. share the first authorship.

## Competing interests

The authors declare no competing interests.
