## [Peer Review File · Nature Communications]

Spatial structure, chemotaxis and quorum sensing shape bacterial biomass accumulation in complex porous mediaEditorial Note: Parts of this Peer Review File have been redacted as indicated to remove third-party material where no permission to publish could be obtained.

Reviewer #1 (Remarks to the Author):

This is a beautiful paper. In it, the authors study how the interplay between fluid flow, cellular motility/chemotaxis, and autoinducer secretion/quorum sensing influences biofilm formation mediated by complex pore geometries, differentiating the influence of dead end pores vs transmitting pores. The experiments are systematically done and carefully analyzed, and provide new insights into this problem, which is of broad interest to researchers in microbiology, biological physics, and studies of transport processes.

The only concerns are regarding the actual presentation of the work, which is confusing in some places, and does not appropriately acknowledge or connect to prior related work, and therefore does not properly contextualize the present results and their importance. As such, the manuscript certainly has potential to be suitable for publication in Nature Communications after some minor revisions are made, as suggested and detailed further below. Congratulations to the authors on performing this nice study.

1. Do the authors observe a similar phenomenon to that reported by <https://doi.org/10.1073/pnas.1300321110>? It doesn't seem so, but why? It is certainly true that the flow rates used in the prior paper are higher, but I am curious about why they authors do not observe an abrupt clogging of the TPs and thus flow disruption after ~ 1 hour. Anyhow, this paper seems extremely relevant to this work and it should be discussed (or at least cited) at some point in the paper. To this point, is there a critical flow rate for the results reported here? I think it would be very interesting to show how sensitive are these experiments to changes in flow rate.

2. A related minor point: are the putative biofilms being formed actually secreting EPS? Or are they simply cellular aggregates? (This is a very minor semantic point, but perhaps worth clarifying explicitly.)

3. Is the Stokes flow simulation necessary? I think measuring the velocity distribution experimentally makes more sense than the simulations given that the authors have access to it. To me it looks like this simulation is shoehorned to have some modeling in the paper. Why do the authors prefer to obtain the velocity field from simulations when could be obtained directly from experiments?

4. To this point, even though it's a considerable effort, the paper would benefit significantly from a minimal model that only considers the three main elements observed in the experiments: bacterial chemotaxis, auto-inducer diffusion and advection, and flow. [Such a model could be a nice extension of <https://doi.org/10.1073/pnas.1300321110>, which is another surprising omission from the present reference list, particularly given the authors' focus on e.g., "The physical structure of the surrounding system controls the spatial and temporal availability of resources and AIs by modulating its local transport." I encourage them to consider discussing its relation to the present work.] The authors can explore if the experiments can be recapitulated when only considering these three components, and thus analyze the predictive capability of the model, which could strengthen the authors' claims and clarify some hypotheses. Also, some quantities that cannot be directly measured in the experiments could be inferred from simulations. Again, this is just a suggestion to help improve the presentation of the results, better situate them within the present literature, and improve the potential impact. I am not proposing the authors perform lots of new experiments or construct a new model for this current paper to be publishable—these ideas could instead be at least addressed and discussed in some form, and could serve as the basis for future work.

5. Another related point: another important and highly relevant paper that is notably not discussed here is <https://doi.org/10.1103/PhysRevLett.129.198102>, which shows that stochastic processes may be important in biofilm formation. I encourage the authors to discuss its relation to the present work.

6. Another notable omission is <https://doi.org/10.1038/349630a0> and <https://doi.org/10.1038/376049a0>, in which the authors showed that starved E coli excrete

another chemoattractant aspartate, which competes with the overall chemotaxis and causes cluster formation. Consider discussing/comparing/contrasting to this work, also the possible influence of secretion of other chemoattractants like aspartate (versus AI2 which is focused on here).

7. It was surprising to read in the introduction that "observations on persistent chemotactic gradients, leading to cell accumulations over several hours - a time scale comparable to the one of bacterial colony growth - have never been reported." I understand that the authors are trying to motivate their findings in DEPs, but this statement is factually untrue and not necessary to motivate the findings. A clear counterexample that I recently came across is <https://doi.org/10.1016/j.bpj.2021.05.012>, which I believe had experiments lasting over ten hours, although there are possibly other examples as well. It is again surprising that the authors did not discuss/compare/contrast to this work, which also experimentally studied chemotaxis in pore geometries.

8. Could the authors visualize bacterial chemotaxis towards the DEPs from time-lapse imaging? It is surprising that they did not include this, given the high quality of their experimental setup.

9. Fig. 1a could be more descriptive and include more information. It would be of great help to describe the different important steps of the main experiments in a visual manner here.

10. There is no reference to Fig. 3b in the Main Text. Is this figure panel necessary given that similar information but at shorter time is plotted in Fig. 3b?

11. Fig. 3d: including labels would help.

12. The central finding that stagnant regions in DEPs promotes biofilm formation is likely also connected to the findings of <https://doi.org/10.1073/pnas.2115496119>, which is also notably not discussed at all.

13. "This is corroborated by biofilms models that incorporate reaction-diffusion effects, where it has been shown that biomass production pushes daughter cells towards a more oxygen rich environment" — it might be useful to mention recent experiments supporting this idea, such as <https://doi.org/10.1073/pnas.2214211120> and <https://www.pnas.org/doi/10.1073/pnas.2208019119>.

14. The authors try to motivate their work by referencing the topology of the human gut, but obviously this study addresses a much simpler problem. It would be useful for them to expand a little more on how their results could connect to the actual gut environment, in which there are other complexities such as e.g., mucus secretion in the intestinal crypts, competition with many other microbial taxa (which are often very concentrated), etc.

Reviewer #2 (Remarks to the Author):

This manuscript describes a multistage accumulation of *E. coli* bacteria within heterogenous microcavities (termed by the authors dead-end pores, or DEPs) in a flow-through microfluidic system. The design of the microfluidic setup is elegant and the observed differences in accumulation between the wildtype *E. coli* and its *d_luxS* derivative are striking. Although the proposed relevance to bacterial ecology in the mammalian gut is very speculative, the results are quite interesting and might indeed have important biological implications.

However, I believe that several key claims of this manuscript are not sufficiently supported by the data:

1) It is concluded that the difference in accumulation in DEPs between the wild type and *luxS*

mutant cells is due to the lack of chemotaxis toward AI-2 in the latter. Although it is a plausible hypothesis, the effects of luxS deletion/ AI-2 in E. coli are known to be pleiotropic (and authors refer to this themselves in lines 272-275). Although the authors confirmed that luxS deletion did not affect motility, it could affect cell adhesion, thus changing dynamics of accumulation, or affect chemotaxis to other compounds beyond AI-2. In order to prove that AI-2 chemotaxis really plays a role in observed accumulation in DEPs, the authors should test a general chemotaxis-deficient strain (e.g., cheY deletion) and lsrB knockout that is specifically deficient in the AI-2 chemotaxis.

2) These controls should also help to clarify whether AI-2 chemotaxis is important during the "intermediate" stage of bacterial accumulation in DEPs. Another control that would be important here is a non-aggregating strain (flu knockout). Of note, authors themselves mention that flu expression might be affected by AI-2 uptake (line 274-275), which would provide an entirely different alternative interpretation for their data than proposed in the manuscript.

3) I do not see any direct evidence that the dense cell aggregates seen for the wild type cells in Figure 5 have anything to do with biofilms. The activation of lsrR reporter is certainly not enough to claim that, since it is not a biofilm marker. Rather, the authors need to show that the expression of reporters for biofilm matrix components (curl, colonic acid, and/or flu) is activated.

4) The rationale for the experiment shown in Figure 3d is not really clear to me. Is it simply to show that gradients of an attractant can persist in the system and attract bacteria? Then it should be explained more clearly. And again, the non-motile strain is not a good control to make this point. The authors should rather use the non-chemotactic but motile cheY deletion strain.

Reviewer #3 (Remarks to the Author):

The manuscript "Spatial structure, chemotaxis and quorum sensing shape biomass accumulation in complex systems" by Scheidweiler et al. reports an experimental study using microfluidics on how the geometry, flow, chemotaxis and quorum sensing together determine the distribution of biomass in porous media. The complex environments of the natural habitats of bacteria motivates the current study. The authors revealed three different stages of biomass growth and provided a detailed explanation/justification of the observed phenomena via genetic engineered bacterial strains and tracking of fluorescent tracers and molecules. The writing of the manuscript is concise and clear. The results of the work are potentially interests to the broad readership of Nat. Commun. Hence, I recommend the publication of the manuscript.

I have a few questions/suggestions, which I hope could be useful for the authors to further improve their manuscript:

1) I am wondering how the geometrical features of the porous systems affect the results. As the motivation of the work, Fig. 1C shows the complex geometry of mice intestine, which shows structures of quite different length scales. The main channel outside large cavities is presumably much wider than the cavities. Within the large cavities, there are many small dead-end cavities too. In comparison, the microfluidic model adopted in the current study has a uniform size for pores, regardless whether they are DEPs or TPs. Suppose if the size of DEPs is much wider than the width of TPs, will the same results still hold at least qualitatively?

2) I am wondering how the motility of bacteria changes over the time. First, in preparation of the system for homogenous filling of bacteria, the time scale is about a couple of hours, do bacteria maintain their motility over this time? Furthermore, with the injection of the sterile M9 minimal medium, how does bacterial motility change over the time of tens of hours?

3) I am curious what is the size of AI-2 molecules compared with that of glucose. The explanation of the results relies on the diffusion of different molecules in and out of DEPs. Do the two molecules have similar diffusion coefficients? Does the difference in their diffusivity play any role in the outcome?

4) Lastly, with the consumption, production and diffusion of glucose and AI-2 all known, it seems that one could write two simple coupled 1D differential equations to solve the concentration profile of the two types of molecules in a dead-ended channel. The results will provide much stronger support to the observed P_{IsrR} activity shown in Fig. 5d than the simple back-of-the-envelope estimate of the diffusion length given in pg. 11.

REVIEWER # 1

“This is a beautiful paper. In it, the authors study how the interplay between fluid flow, cellular motility/chemotaxis, and autoinducer secretion/quorum sensing influences biofilm formation mediated by complex pore geometries, differentiating the influence of dead end pores vs transmitting pores. The experiments are systematically done and carefully analyzed, and provide new insights into this problem, which is of broad interest to researchers in microbiology, biological physics, and studies of transport processes.

The only concerns are regarding the actual presentation of the work, which is confusing in some places, and does not appropriately acknowledge or connect to prior related work, and therefore does not properly contextualize the present results and their importance. As such, the manuscript certainly has potential to be suitable for publication in Nature Communications after some minor revisions are made, as suggested and detailed further below. Congratulations to the authors on performing this nice study.”

We thank the reviewer for their careful reading of the paper and the valuable comments that have helped improve the manuscript. We also thank the reviewer for a globally very positive assessment of this work. Below, we address a detailed point by point response to all the remarks that have been raised. We also point out how we revised the paper in accordance with the reviewer’s suggestions. The changes and additions are highlighted with blue colored texts in the revised manuscript.

“1. Do the authors observe a similar phenomenon to that reported by <https://doi.org/10.1073/pnas.1300321110>? It doesn’t seem so, but why? It is certainly true that the flow rates used in the prior paper are higher, but I am curious about why they authors do not observe an abrupt clogging of the TPs and thus flow disruption after ~ 1 hour. Anyhow, this paper seems extremely relevant to this work and it should be discussed (or at least cited) at some point in the paper. To this point, is there a critical flow rate for the results reported here? I think it would be very interesting to show how sensitive are these experiments to changes in flow rate.”

We do not observe a similar phenomenon as the one reported in the mentioned paper (Drecher et al. PNAS 2013). There, the flow-study is conducted with imposed pressure with *P. aeruginosa* strain while we work under imposed flow with *E. coli* strain. On the one hand our microbial strain has different outer surface properties than the one used in Drecher et al. PNAS 2013. On the other hand, imposed flow and imposed pressure are two very different mechanical conditions under which fluid motion can be established. For imposed external pressure drop between inlet and outlet, as used in the mentioned work, the flow adapts to the medium variable structural properties: if, due to microbial accumulation and growth, the local pore size gets reduced, also flow and shear locally reduce as the pressure stays constant. Under these conditions the accumulated mass tends to stays in place and keeps accumulating. In our case, we impose the overall flow, thus, the pressure adapts to the local structural conditions: if, due to microbial accumulation and growth, the local pore size gets reduced, the pressure increases as the flow stays constant and the shear stress exerted by the fluid on the growing biomass piles up and eventually removes colonies (or part of them). This is what happens in our microfluidics system: after an initial transient, the biomass growing and accumulating in the transmitting pores (TP) finds an equilibrium between growth by cell division and removal due to fluid shear. The reviewer rises an important point and we decided to clarify that in the revised version of the manuscript. In particular, we added the following sentence at line 154 of the revised manuscript:

In previous studies the biomass growth under macroscopic imposed pressure conditions reported the catastrophic disruption of the flow as consequence of the reduction of the space available to fluid flow due to cellular division [ref]. Here, as we impose a constant flow rate, we do not observe such phenomenon.

The reviewer also wonders how would our results change for different flow rates. While this lies outside the scope of our contribution, we believe that it is a good point. We argue that our results do not depend on the specific value of the imposed flow rate Q , as far as it is strong enough to re-fresh nutrients and oxygen while

depleted in the Transmitting Pores but it is low enough to be within the laminar regime. As far as a system is composed of percolating channels (the TP) of similar size connecting DEPs of heterogeneously distributed depth, then: i) the biomass accumulating in the TP will reach a carrying capacity controlled by the interplay between nutrients/oxygen concentration and fluid shear and ii) the biomass accumulating in the DEP is controlled by chemotaxis at earlier times and quorum sensing at later ones. In particular, we added the following sentence at the end of the results section, at line 443 of the revised manuscript:

We argue that our results do not depend on the specific value of the imposed flow rate Q , as far as it is strong enough to re-fresh nutrients and oxygen while depleted in the Transmitting Pores but it is low enough to be within the laminar regime.

2. A related minor point: are the putative biofilms being formed actually secreting EPS? Or are they simply cellular aggregates? (This is a very minor semantic point, but perhaps worth clarifying explicitly.)

We do not have data answering this question. The observed biomass is growing by cell divisions taking place within the microfluidics chamber (as we inject sterile nutrients), but daughter cells could move, swimming or passively transported, towards new locations contributing to other colonies/aggregates. Therefore, we modify the text to substitute the word biofilms with colonies/aggregates.

3. Is the Stokes flow simulation necessary? I think measuring the velocity distribution experimentally makes more sense than the simulations given that the authors have access to it. To me it looks like this simulation is shoehorned to have some modeling in the paper. Why do the authors prefer to obtain the velocity field from simulations when could be obtained directly from experiments?

We decided to show the magnitude of the flow field to help the reader to familiarize with the physical transport conditions of the experiments. As it has been shown for a similar structure in *Bordoloi and Scheidweiler et al. Nat Comm. 2022*, the flow field within DEP develops laminar vortexes of decaying intensity spanning across several orders of magnitude. Thus, PIV experimental methods cannot resolve such complex flow structures. We include in the revised manuscript supporting information the PIV measurements we did showing that the flow field is properly resolved only within TP and not within DEP. Thus, we included the flow simulations to display the flow structure emerging in such locations.

4. To this point, even though it's a considerable effort, the paper would benefit significantly from a minimal model that only considers the three main elements observed in the experiments: bacterial chemotaxis, auto-inducer diffusion and advection, and flow. [Such a model could be a nice extension of <https://doi.org/10.1073/pnas.1300321110>, which is another surprising omission from the present reference list, particularly given the authors' focus on e.g., "The physical structure of the surrounding system controls the spatial and temporal availability of resources and AIs by modulating its local transport." I encourage them to consider discussing its relation to the present work.] The authors can explore if the experiments can be recapitulated when only considering these three components, and thus analyze the predictive capability of the model, which could strengthen the authors' claims and clarify some hypotheses. Also, some quantities that cannot be directly measured in the experiments could be inferred from simulations. Again, this is just a suggestion to help improve the presentation of the results, better situate them within the present literature, and improve the potential impact. I am not proposing the authors perform lots of new experiments or construct a new model for this current paper to be publishable—these ideas could instead be at least addressed and discussed in some form, and could serve as the basis for future work.

The reference suggested is the same one of the previous point #1. The model proposed in that reference is for the overall medium permeability that changes while bacterial cells grow in number occupying more and more space, reducing the one available for the fluid to flow. This issue is probably resolved with the clarification we provided in point #1, as we do use imposed flow and not pressure, thus, the medium intrinsic permeability does not change much with accumulating biomass, in time. Following this reviewer suggestion, we developed a numerical model (described in the methods section) to couple the i) diffusion and bacterial consumption of Glucose to ii) diffusion, production and uptake of AI-2 within DEP to predict the longitudinal signal of the *IsrR* reported along the DEP. The numerical model results, shown in figure S6, agree well with our theoretical prediction of the location ξ where a sharp increase of the *IsrR* activity is observed, as result of glucose depletion and AI-2 internalization.

5. Another related point: another important and highly relevant paper that is notably not discussed here is <https://doi.org/10.1103/PhysRevLett.129.198102>, which shows that stochastic processes may be important in biofilm formation. I encourage the authors to discuss its relation to the present work.

This is indeed an interesting model for Quorum Sensing (QS) and its impact on growing biofilm on smooth and solid surfaces in absence of flow. The authors present a macroscopic model for QS biofilm growth, where a transition of bacterial cells to their biofilm state takes place as the cellular density overcomes a critical threshold. While model accounts for immigration, detachment, proliferation and migration of bacterial cells within the biofilm it does not account for flow and nutrients limitations. We now cite that reference and, as suggested, we discuss its relation to our present work.

6. Another notable omission is <https://doi.org/10.1038/349630a0> and <https://doi.org/10.1038/376049a0>, in which the authors showed that starved *E. coli* excrete another chemoattractant aspartate, which competes with the overall chemotaxis and causes cluster formation. Consider discussing/comparing/contrasting to this work, also the possible influence of secretion of other chemoattractants like aspartate (versus AI2 which is focused on here).

We thank the reviewer for rising this point. Of course, *E. coli* can produce aspartate under starvation. In our case, we believe that this does not take place at early time, when cells are exposed to nutrients, but rather it is plausible to happens at later times, deep inside the DEP where nutrient refreshing is limited by diffusion while consumed by the cells accumulated there. We now cite these references and we discuss that at late times, within the DEP where cells can be under starvation, other sources of chemotaxis cannot be excluded. We, however, believe that was not our case. Being the $\Delta luxS$ mutant capable of chemotaxis towards any other compounds, at least for the experiments with that mutant we do not observe biomass accumulation that can indicate a chemotactic migration.

Line 363 pdf:

These findings complement previous works, where has been shown that in absence of flow *E. coli* exhibits a chemotactic response towards gradients of self-secreted amino acids [57–59], or while consuming resources, migrating towards areas with higher concentration [33]. Additionally, not only concentration, but also quality of carbon may influence the bacterial chemotactic response. In *E. coli*, the expression of motility genes is inversely proportional to the growth rate achieved on different carbon sources [60]. In most conditions, glucose is the best carbon source for *E. coli*, and for this study, we chose a concentration comparable with that found in mammalian guts (5-50 mM) [45]. In presence of other resources, chemotaxis could therefore result in different colonization pattern.

7. It was surprising to read in the introduction that “observations on persistent chemotactic gradients, leading to cell accumulations over several hours - a time scale comparable to the one of bacterial colony growth - have never been reported.” I understand that the authors are trying to motivate their findings in DEPs, but this statement is factually untrue and not necessary to motivate the findings. A clear counterexample that I recently came across is <https://doi.org/10.1016/j.bpj.2021.05.012>, which I believe had experiments lasting over ten hours, although there are possibly other examples as well. It is again surprising that the authors did not discuss/compare/contrast to this work, which also experimentally studied chemotaxis in pore geometries.

We agree with the reviewer and we now cite the reference suggested and we discuss it with respect to our work. The suggested reference is about a study of bacterial spatial distribution as consequence of chemotaxis along gradients produced by the cells themselves by consuming nutrient which is the chemoattractant itself. In that case there is no flow, in our case the chemotactic gradient is sustained by the medium structure, flow and the chemoattractant production by resident cells, and experienced by the incoming ones that get attracted towards DEP. Thus, we modified that sentence in the introduction (line 67), as follows:

Observations on persistent chemotactic gradients, maintained in porous medium over several hours have been reported in the absence of flow when bacteria consume nutrient that is a chemoattractant itself [33]. Here we consider the scenario where the medium structure, together with bacteria transport, aggregation and chemoattractant production sustain gradients persistently in presence of flow.

8. *Could the authors visualize bacterial chemotaxis towards the DEPs from time-lapse imaging? It is surprising that they did not include this, given the high quality of their experimental setup.*

A control experiment of bacterial transport over about 2 hours with and without a chemoattractant (Serine) is also shown in figure 3d. We also report in figure 3e the trajectories (time-lapse) of chemotactic cells towards the gradient of serine in a DEP. Moreover, we do observe the consequent accumulation of bacterial cells, as reported qualitatively in figure 3a and quantitatively in figure 3b. We modified the figure 3 caption accordingly.

...(E) Gradient of the fluorescent tracer (top panel) and trajectories of *E. coli* WT (from time-lapse imaging) attracted towards a chemoattractant, after 1 PV (24 min). Scale bar 50 μm .

9. *Fig. 1a could be more descriptive and include more information. It would be of great help to describe the different important steps of the main experiments in a visual manner here.*

Following the reviewer comment, we modified figure 1 adding a schematic (now figure 1b) summarizing the important steps of the main experiments in a visual manner.

10. *There is no reference to Fig. 3b in the Main Text. Is this figure panel necessary given that similar information but at shorter time is plotted in Fig. 3b?*

In the revised manuscript we refer to all subfigures and plots presented in all figures. After careful revision, we believe that all figures are necessary and help to convey our message.

11. Fig. 3d: including labels would help.

Following this reviewer suggestion, we modified the figure 3d accordingly, adding labels.

12. *The central finding that stagnant regions in DEPs promotes biofilm formation is likely also connected to the findings of <https://doi.org/10.1073/pnas.2115496119>, which is also notably not discussed at all.*

We agree with the reviewer and we thank they for pointing us towards this reference that is now cited and discussed.

13. *"This is corroborated by biofilms models that incorporate reaction-diffusion effects, where it has been shown that biomass production pushes daughter cells towards a more oxygen rich environment" — it might be useful to mention recent experiments supporting this idea, such as <https://doi.org/10.1073/pnas.2214211120> and <https://www.pnas.org/doi/10.1073/pnas.2208019119>.*

We agree with the reviewer and we thank they for pointing this out. We now mention these recent experimental findings that support the idea we propose.

14. *The authors try to motivate their work by referencing the topology of the human gut, but obviously this study addresses a much simpler problem. It would be useful for them to expand a little more on how their results could connect to the actual gut environment, in which there are other complexities such as e.g., mucus secretion in the intestinal crypts, competition with many other microbial taxa (which are often very concentrated), etc.*

We now expanded the results section of our work discussing how the simplified system we consider is useful to understand also much more complex systems (as the actual gut environment), where other sources of complexity (including mucus secretion in the intestinal crypts, competition with other taxa for nutrients and space, etc.) takes place simultaneously.

REVIEWER #2

This manuscript describes a multistage accumulation of E. coli bacteria within heterogenous microcavities (termed by the authors dead-end pores, or DEPs) in a flow-through microfluidic system. The design of the microfluidic setup is elegant and the observed differences in accumulation between the wildtype E. coli and its $\Delta luxS$ derivative are striking. Although the proposed relevance to bacterial ecology in the mammalian gut is very speculative, the results are quite interesting and might indeed have important biological implications.

We thank the reviewer for their careful reading of the paper and the valuable comments that have helped improve the manuscript. We also thank the reviewer for a globally positive assessment of this work that have important biological implications. Below, we address a detailed point by point response to all the remarks that have been raised. We also point out how we revised the paper in accordance with the reviewer's suggestions. The changes and additions are highlighted with blue colored texts in the revised manuscript.

However, I believe that several key claims of this manuscript are not sufficiently supported by the data:

1) It is concluded that the difference in accumulation in DEPs between the wild type and luxS mutant cells is due to the lack of chemotaxis toward AI-2 in the latter. Although it is a plausible hypothesis, the effects of luxS deletion/ AI-2 in E. coli are known to be pleiotropic (and authors refer to this themselves in lines 272-275). Although the authors confirmed that luxS deletion did not affect motility, it could affect cell adhesion, thus changing dynamics of accumulation, or affect chemotaxis to other compounds beyond AI-2. In order to prove that AI-2 chemotaxis really plays a role in observed accumulation in DEPs, the authors should test a general chemotaxis-deficient strain (e.g., cheY deletion) and lsrB knockout that is specifically deficient in the AI-2 chemotaxis.

We understand that the main conclusion of our work has been misinterpreted: the revised manuscript has been improved to avoid that. We do not suggest that the difference in the observed spatial biomass accumulation between the two strains, is due to a lack of chemotaxis of $\Delta luxS$ mutant. We modified the manuscript in order to clarify our statements and avoid such misinterpretation. Our results show that the presence of AI-2 gradients attract chemotactically the wild type *E. coli* towards the DEPs, giving rise to a non-homogenous spatial organization of the biomass, that get amplified by cell division and QS. Differently, the $\Delta luxS$ mutant exhibits a uniform spatial organization, not because lacks chemotaxis, but because the chemoattractant AI-2 is missing in the environment.

The reviewer has concerns about the choice of the $\Delta luxS$ mutant used in this study, hypothesizing that the removal of the gene *luxS* could affect cell adhesion or chemotaxis towards other compounds, thus impacting dynamics of accumulation. In this regard, we would like to clarify that our experimental approach aims to compare a scenario where *E. coli* wild type cells release extracellularly AI-2, to a scenario where no AI-2 is produced. The coupling between the heterogeneous flow and the extracellular release of AI-2 by the cells induces and sustains the presence of AI-2 gradients. The interplay between topological features of the solid structure, flow, and biology results in higher AI-2 concentration within the DEPs where fluid is stagnant, and low concentration in the TPs where the flow of fresh medium dilutes and advects the AI-2 molecules. To the best of our knowledge, the only possibility to achieve this goal is by using a $\Delta luxS$ isogenic mutant. In addition, we chose the $\Delta luxS$ mutant because it is known from previous work that *luxS* does not affect chemotaxis towards AI-2, see fig 1 (Jani et al., 2017), nor swimming speed, as confirmed by our experiments (fig S1).

Redacted

Fig1. Assessment of chemotaxis towards AI-2 for different *E. coli* knockout mutants (Jani et al., 2017)

We show that at later times *E. coli* $\Delta luxS$ is capable of forming aggregates (fig 4a,b). However, after 20 hrs, *E. coli* $\Delta luxS$ aggregates stop increasing in size, while the wild type exhibits an increase of the aggregate size until occupying the entire DEP space. We argue that this differentiation in aggregation dynamics results from the collision encounter rate, which is enhanced by chemotactic migration in the WT. We modify the text for clarity.

Line 262: This is captured in the aggregate-averaged mass distribution (probability density function, PDF) within the DEP that stays nearly constant for $\Delta luxS$ but keeps shifting towards larger values for WT (Fig 4b), thus implying that the aggregate size increase in DEPs stems from a combination of binary cell division and recruitment of new cells via chemotaxis.

Please note that it has been previously shown that aggregation in *E. coli* is only partially reduced for $\Delta cheY$ and $\Delta lsrB$ (Laganenka et al., 2016; Laganenka and Sourjik, 2018), see figure 2. The options suggested by the reviewer ($\Delta cheY$ or $\Delta lsrB$), while being deficient in chemotaxis, would still produce AI-2, whose presence could in turn influence QS and therefore further complicating the data interpretation.

Redacted

Fig2. Aggregation dynamics for different *E. coli* knockout mutants (Laganenka et al., 2016).

While the presence of AI-2 is the only difference between our experiments with WT, showing accumulation in DEP, and the control with $\Delta luxS$ mutant, that accumulate homogeneously throughout the whole system, we cannot exclude that chemotaxis towards other compounds could happen. We therefore modify the text accordingly:

Line 368 pdf:

These findings complement previous works, where has been shown that in absence of flow *E. coli* exhibits a chemotactic response towards gradients of self-secreted amino acids [57–59], or while consuming resources, migrating towards areas with higher concentration [33]. Additionally, not only concentration, but also quality of carbon may influence the bacterial chemotactic response. In *E. coli*, the expression of motility genes is inversely proportional to the growth rate achieved on different carbon sources [60]. In most conditions,

glucose is the best carbon source for *E. coli*, and for this study, we chose a concentration comparable with that found in mammalian guts (5-50 mM) [45]. In presence of other resources, chemotaxis could therefore result in different colonization pattern.

2) These controls should also help to clarify whether AI-2 chemotaxis is important during the "intermediate" stage of bacterial accumulation in DEPs. Another control that would be important here is a non-aggregating strain (*flu* knockout). Of note, authors themselves mention that *flu* expression might be affected by AI-2 uptake (line 274-275), which would provide an entirely different alternative interpretation for their data than proposed in the manuscript.

First, the reviewer suggests to use a *flu* knockout mutant. However, like antigen 43 (*Ag43*), the product of the *flu* gene, other adhesins (i.e. Va autotransported adhesins TibA or AIDA-I (SAATs), trimeric autotransporters), conjugation pili, or fimbriae including curli or type IV pilus, can affect aggregation (Chekli et al., 2023). Understanding which protein is involved in the aggregation that we report would certainly be a very insightful information, however it is not the objective of the presented research. This work has not been designed to investigate the molecular mechanism of aggregation, for which a different experimental approach should have been considered. Here, the goal is to understand how heterogeneous flow generates gradients of the signaling molecule AI-2, and how these gradients control biomass spatial distribution of *E. coli* in our model system. Resolving whether the aggregation is controlled by the adhesin *Ag43*, pili, curli fibers, or EPS is out of the scope of this work. Detailed investigation of chemotaxis towards AI-2 and the aggregation of *E. coli* has been previously covered by other studies (Laganenka et al., 2016; Laganenka and Sourjik, 2018), now cited in our manuscript, that addressed the role of Δ *LsrB*, Δ *cheY* and Δ *flu* on the AI-2 mediated aggregation.

Second, it is known that LsrR controls AI-2 uptake and regulates the expression of several biofilm-related genes such as *wza*, involved in the synthesis of colanic acid a building block of EPS, and *flu*, which encodes the proteins that mediates cell-to-cell aggregation. However, this would occur only after glucose has been depleted below a concentration which allows AI-2 internalization. This is the reason why we differentiate between the three phases: chemotaxis and accumulation of individual cells, aggregation mediated by collision and binary growth, and later glucose depletion and QS driven biomass enhanced growth. Thus, even considering the interaction between AI-2 uptake (late times) and aggregation mechanism (happening at early/intermediate times) will not provide a different interpretation of our results.

3) I do not see any direct evidence that the dense cell aggregates seen for the wild type cells in Figure 5 have anything to do with biofilms. The activation of *LsrR* reporter is certainly not enough to claim that, since it is not a biofilm marker. Rather, the authors need to show that the expression of reporters for biofilm matrix components (*curl*, colonic acid, and/or *flu*) is activated.

We agree that the observation of the *LsrR* reporter signal is not evidence for biofilm formation, which is not our claim. To avoid any confusion, we removed the expression "biofilm" from our manuscript. Here, the reporter has been used as an evidence for AI-2 import and QS activation, as we now clarify in the text:

L287

To test if the observed biomass accumulation is the result of QS, we estimate from the activity of the *LsrR* transcriptional reporter, when and where *E. coli* cells start to internalize AI-2, and if there is a temporal agreement with the quantified biomass accumulation dynamics.

4) The rationale for the experiment shown in Figure 3d is not really clear to me. Is it simply to show that gradients of an attractant can persist in the system and attract bacteria? Then it should be explained more clearly. And again, the non-motile strain is not a good control to make this point. The authors should rather use the non-chemotactic but motile *cheY* deletion strain.

The rationale behind the transport experiments, resumed in figure 3d, is to show that ephemeral gradients (that last for short time) within the small space of a pore (i.e. around 30 microns) are sufficient to trigger a chemotactic accumulation of *E. coli* WT in the DEPs. For this reason, it is not appropriate here to compare the effect of chemotaxis with a non-chemotactic mutant (both WT and the *luxS* deficient mutant considered here can do chemotaxis). Thus, the biological control is represented by the experiment without serine. The non-motile strain has been used as a passive tracer, as particles that are simply transported by the flow and cannot swim randomly

or towards any chemical gradient. All the transport experiments have been performed by mixing motile and non-motile cells, tagged by different fluorophores. Therefore, the behavior of the passive tracers was considered as a proof of technical replicability between experiments. We now better clarify the purpose of these transport experiments at line 224:

In a separate transport experiment, we examine if these small scale and persistent gradients are sufficient to trigger the chemotactic motion by *E. coli* cells.

REVIEWER #3

The manuscript "Spatial structure, chemotaxis and quorum sensing shape biomass accumulation in complex systems" by Scheidweiler et al. reports an experimental study using microfluidics on how the geometry, flow, chemotaxis and quorum sensing together determine the distribution of biomass in porous media. The complex environments of the natural habitats of bacteria motivates the current study. The authors revealed three different stages of biomass growth and provided a detailed explanation/justification of the observed phenomena via genetic engineered bacterial strains and tracking of fluorescent tracers and molecules. The writing of the manuscript is concise and clear. The results of the work are potentially interests to the broad readership of Nat. Commun. Hence, I recommend the publication of the manuscript.

We thank the reviewer for their careful reading of the paper and the valuable comments that have helped improve the manuscript. We also thank the reviewer for a globally very positive assessment of this work that they suggest to publish. Below, we address a detailed point by point response to all the remarks that have been raised. We also point out how we revised the paper in accordance with the reviewer's suggestions. The changes and additions are highlighted with blue colored texts in the revised manuscript.

I have a few questions/suggestions, which I hope could be useful for the authors to further improve their manuscript:

1) I am wondering how the geometrical features of the porous systems affect the results. As the motivation of the work, Fig. 1C shows the complex geometry of mice intestine, which shows structures of quite different length scales. The main channel outside large cavities is presumably much wider than the cavities. Within the large cavities, there are many small dead-end cavities too. In comparison, the microfluidic model adopted in the current study has a uniform size for pores, regardless whether they are DEPs or TPs. Suppose if the size of DEPs is much wider than the width of TPs, will the same results still hold at least qualitatively?

This is an interesting point. We argue that the structural feature that control the observed phenomena here is the structural duality: TP vs DEP. As far as a system is composed of percolating channels of similar size (the TP) connecting DEPs of heterogeneously distributed depth, then: i) the biomass accumulating in the TP will reach a carrying capacity controlled by the interplay between nutrients/oxygen concentration and fluid shear and ii) the biomass accumulating in the DEP is controlled by chemotaxis at earlier times and quorum sensing at later ones. We argue that, overall, these processes should happen also for TP of distributed size and DEP of distributed width (and not only depth). In the first case, the flow and transport of nutrients and oxygen in each the TPs should be controlled by their own width w , as the local permeability is expected to vary as w^2 . In the second case, let's remind that we define DEP as the structural unit of the medium for which the maximum inscribed circle touches the same grain in three points: this happens when the cavity within the grain is deeper than wider. In this situation, as discussed in Bordoloi and Scheidweiler et al. Nat Comm. 2022, the fluid flow exhibits complex laminar vortexes of exponentially decaying intensity moving towards the DEP. Thus, we expect that even for width-distributed DEP, as far as they are DEP according to our geometrical definition, the fluid flow within them is so low to sustain the bacterial accumulation and gradient persistence at the DEP-TP juncture and, finally, the late time QS-dominated biomass production. We clarify this point in the revised manuscript, as follows:

We argue that, overall, the impact of medium structure, chemotaxis and quorum sensing on bacterial biomass accumulation described here takes place also for TPs of distributed size and DEP of distributed width (and not only depth). In the first case, the flow and transport of nutrients and oxygen in each the TPs should be controlled by their own width w , as the local permeability is expected to vary as w^2 (as in a pipe). For the second case, let's consider that we defined a DEP as the structural unit of the medium for which the local maximum inscribed circle touches the solid grain structure in three points belonging to the same grain. This condition happens when the cavity within a grain is deeper than wider. In this situation, as discussed in Bordoloi and Scheidweiler et al. Nat Comm. 2022, the fluid flow exhibits complex laminar vortexes of exponentially decaying intensity moving inside the DEP. Thus, we expect that even for width-distributed DEP, as far as they are DEP according to our geometrical definition, the fluid flow within them is so low to sustain the bacterial accumulation and gradient persistence at the DEP-TP juncture and, finally, the late time QS-dominated biomass production.

2) I am wondering how the motility of bacteria changes over the time. First, in preparation of the system for homogenous filling of bacteria, the time scale is about a couple of hours, do bacteria maintain their motility

over this time? Furthermore, with the injection of the sterile M9 minimal medium, how does bacterial motility change over the time of tens of hours?

During the bacterial injection phase, cells were suspended in motility buffer that ensure their motility for more than the two hours of the injection (as we verified also with no flow experiments used to measure their motility by particle tracking). As we switch to sterile nutrient injection, the suspended cells are a mixture of cells from recent divisions and older ones. Anyway, within the TP the concentration of oxygen and nutrients is supported by their constant injection: under these conditions the cells can swim propelled by their flagella. We included in the new submission a brief discussion of this in the supplementary information.

3) I am curious what is the size of AI-2 molecules compared with that of glucose. The explanation of the results replies on the diffusion of different molecules in and out of DEPs. Do the two molecules have similar diffusion coefficients? Does the difference in their diffusivity play any role in the outcome?

The molecular mass of Glucose is about 192 g/mol and the one of the Autoinducer-2 is about 180 g/mol: thus, their diffusion coefficient are reasonably of the same order of magnitude. We include this information in the new submission and we use the (not identical) diffusion coefficients computed with the Stokes-Einstein formula in the numerical models (see point #4).

4) Lastly, with the consumption, production and diffusion of glucose and AI-2 all known, it seems that one could write two simple coupled 1D differential equations to solve the concentration profile of the two types of molecules in a dead-ended channel. The results will provide much stronger support to the observed P_{LsrR} activity shown in Fig. 5d than the simple back-of-the-envelope estimate of the diffusion length given in pg. 11.

We agree and we thank the reviewer for this very nice suggestion. We developed a 1-dimensional numerical model to simulate diffusion and consumption of glucose and diffusion, production and uptake of AI-2 along a single DEP. We included a full description of that numerical model in the methods section and we included the results in figure S6. The results of the diffusion-reaction numerical simulation support the prediction of the LsrR signal spatial variation position, ξ .

- Chekli, Y., Stevick, R.J., Kornobis, E., Briolat, V., Ghigo, J.-M., Beloin, C., 2023. Escherichia coli Aggregates Mediated by Native or Synthetic Adhesins Exhibit Both Core and Adhesin-Specific Transcriptional Responses. *Microbiology Spectrum* 0, e00690-23. <https://doi.org/10.1128/spectrum.00690-23>
- Jani, S., Seely, A.L., Peabody V, G.L., Jayaraman, A., Manson, M.D., 2017. Chemotaxis to self-generated AI-2 promotes biofilm formation in Escherichia coli. *Microbiology* 163, 1778–1790. <https://doi.org/10.1099/mic.0.000567>
- Laganenka, L., Colin, R., Sourjik, V., 2016. Chemotaxis towards autoinducer 2 mediates autoaggregation in Escherichia coli. *Nat Commun* 7. <https://doi.org/10.1038/ncomms12984>
- Laganenka, L., Sourjik, V., 2018. Autoinducer 2-Dependent Escherichia coli Biofilm Formation Is Enhanced in a Dual-Species Coculture. *Appl Environ Microbiol* 84. <https://doi.org/10.1128/AEM.02638-17>

Reviewer #1 (Remarks to the Author):

The authors have suitably addressed the previous reviews, and the manuscript is now suitable for publication.

Reviewer #2 (Remarks to the Author):

I cannot say that I am satisfied with the authors' replies to my comments. While some of the minor comments have been addressed in the revised version of the manuscript, my major concern remains.

To repeat it again: In order to conclusively claim that the observed *E. coli* accumulation in the cavities is due to the chemotaxis toward AI-2, it is simply not sufficient to have a *d_luxS* strain not producing AI-2 as a control. Authors must also include a general non-chemotactic *E. coli* strain (e.g., *d_cheY* or *d_cheA*) and an AI-2 receptor-deficient strain (either *d_lsrB* or *d_tsr*) as further negative controls. These controls would be expected in any solid publication in the bacterial chemotaxis field. Without them, I cannot honestly recommend this manuscript for publication.

Reviewer #3 (Remarks to the Author):

I thank the authors for answering my questions. I am very satisfied with the answers. Thus, I recommend the publication of the work as is.

Reviewer 2

We thank the reviewer for his/her careful reading of the paper and its revision, for the valuable comments that have helped improve the manuscript with two new control experiments. Below, we address the issue that was raised. We also point out how we revised the paper and the supporting information in accordance with the reviewer's suggestions. The modifications are quoted below and are highlighted in the track changes file.

To repeat it again: In order to conclusively claim that the observed *E. coli* accumulation in the cavities is due to the chemotaxis toward AI-2, it is simply not sufficient to have a $\Delta luxS$ strain not producing AI-2 as a control. Authors must also include a general non-chemotactic *E. coli* strain (e.g., $\Delta cheY$ or $\Delta cheA$) and an AI-2 receptor-deficient strain (either $\Delta lsrB$ or Δtsr) as further negative controls. These controls would be expected in any solid publication in the bacterial chemotaxis field. Without them, I cannot honestly recommend this manuscript for publication.

We followed the reviewer suggestions performing new control experiments to address this point in full. The revised manuscript includes novel experiments with the strains suggested by the reviewer. To do that, we first modified our WT *E. coli* MG 1655 strain deleting i) *cheA* gene to get a general non-chemotactic mutant and deleting ii) *tsr* gene to get an AI-2 receptor deficient mutant. Then, we performed flow experiments in the same conditions as for the other strains (WT, $\Delta luxS$ mutant and *lsrR* reporter strains): once the porous system is saturated with the prepared bacterial suspension (overnight, re-grown for 3 hours, washed and re-suspended in motility buffer), we displaced it with sterile M9 solution amended with Glucose. Both flow experiment controls (with mutant strains deficient Δtsr and $\Delta cheA$) have been separately reproduced in triplicate.

The flow experiments have been analyzed as the others. The analysis results support our previous conclusions: in absence of chemotaxis towards AI₂, cells do not significantly accumulate in DEP more than in TP. Thus, the accumulation of WT *E. coli* in Dead-End Pores (at late times 6 times more than in transmitting pores, TP) is due to chemotaxis towards self-produced AI-2, leading to the observed late time quorum sensing. The new data and figures are shown in the Supporting Information. While we kept the current manuscript organization based on the description of the two WT and $\Delta luxS$ strains, we modified the discussion section at line 363-367 as follows.

To conclusively claim that the observed *E. coli* accumulation in the DEP is due to the chemotactic activity toward the self-produced AI-2, we also consider the general non-chemotactic *E. coli* strain $\Delta cheA$ and the AI-2 receptor-deficient strain Δtsr , as further negative controls. Flow experiments performed in triplicates with these two strains, separately, display similar results as the ones described for $\Delta luxS$, in terms of early times DEP-TP individual cells accumulation (see Fig. S5 a and b) and overall biomass accumulation (see Fig. S6 a and b).

The figures added to the supplementary information (Fig. S5 and Fig. S6) quantifying these novel observations, that strengthen our conclusions, are reported below. We believe that these revisions reply in full to the issue raised by the reviewer and substantiate our novel finding on the interplay between pore-scale structure, solute gradients distribution, chemotaxis and quorum sensing, to control the overall biomass growth and spatial organization.

Figure S5. Transport mediated accumulation of *E. coli* at early times (0-12 hours). (A) Images at different times representing bacterial cells abundance in a single DEP, highlighted in red, and the nearest TP, highlighted in cyan, for *E. coli* WT, *E. coli* $\Delta luxS$, *E. coli* $\Delta cheA$ and *E. coli* Δtsr (from top to bottom). Scale bar 50 μm . (B) Retention curves representing the ratio between the concentration of bacteria in the dead-end pores (C_{DEP}) and their concentration in the transmitting pores (C_{TP}) at different times, for all the four strains mentioned above.

Figure S6. Macroscopic biomass accumulation, *E. coli* WT, *E. coli* Δ luxS, *E. coli* Δ cheA and *E. coli* Δ tsr (from top to bottom). (A) Overview of microfluidic chip depicting biased biomass accumulation through the microfluidic geometry at 20, 30, 40 ad 50 hours since nutrient injection. Scale bar 200 μ m. (B) Temporal evolution of the biomass concentration in the TP (cyan), in the DEP (red).